# LLM2Prune: Using LLMs as Domain Experts for Search Space Reduction

**Ankur Nath**  *anath@tamu.edu*
*Department of Computer Science & Engineering*
*Texas A&M University*

**Alan Kuhnle**  *kuhnle@tamu.edu*
*Department of Computer Science & Engineering*
*Texas A&M University*

**Reviewed on OpenReview:** *https://openreview.net/forum?id=i56Pxr3btq*

## Abstract

Combinatorial optimization problems defined over graphs involve large discrete search spaces where many candidates contribute little due to redundancy or low value. Pruning the ground set to a smaller pool of promising candidates makes heuristics and exact solvers practical for large real-world instances. Classical submodularity-based pruning algorithms do not scale efficiently, while learning-based approaches depend on handcrafted features that require domain expertise and limit generalization. We propose LLM2PRUNE, a framework that uses large language models (LLMs) to generate features from a task description, which are then used by a downstream classifier to prune the search space. We guide the feature discovery process with feature-importance scores and performance metrics. Across diverse graph optimization tasks, LLM2PRUNE prunes over 90% of the ground set while retaining near-optimal solutions, achieving orders-of-magnitude speedups over existing approaches. Code, data, and pre-trained models are available at: `https://github.com/ankurnath/LLM_Pruning.git`.

## 1 Introduction

Combinatorial optimization problems on graphs arise in many application domains, including social networks, transportation, telecommunications, and scheduling, and are typically NP-hard. For instance, in facility location problems such as opening retail outlets under a budget constraint, many candidate sites are unpromising due to high costs, low demand, or redundancy with nearby locations. In fact, among the 21 NP-complete problems introduced by Karp (Karp, 2009), 10 are decision versions of graph optimization problems, while most of the remaining problems, such as set covering, can be naturally formulated on graphs. Several heuristics have been developed to obtain near-optimal solutions, but they do not typically scale well with problem size. Pruning the ground set to a smaller pool of promising candidates enables heuristics and exact methods to scale more efficiently without sacrificing solution quality.

In this work, we study the problem of identifying a smaller subset of the ground set for graph optimization problems, where the goal is to maximize or minimize an objective function over a large discrete configuration space subject to constraints. Formally, we have the following problem definition for the general CO problem.

**Problem Definition (Pruning).** A CO problem can be expressed as the tuple $(S, \Omega, f)$, where $S$ denotes the search space over discrete decision variables, $\Omega$ represents the set of constraints that define feasibility, and $f : S \to \mathbb{R}$ is the objective function to be optimized. Given a CO problem with ground set $\mathcal{U}$ and an exact solver $\mathcal{OPT}$, the optimal solution is

$$X^\star = \mathcal{OPT}(\mathcal{U}) = \arg \max_{X \subseteq \mathcal{U}, \ X \text{ satisfies } \Omega} f(X)$$

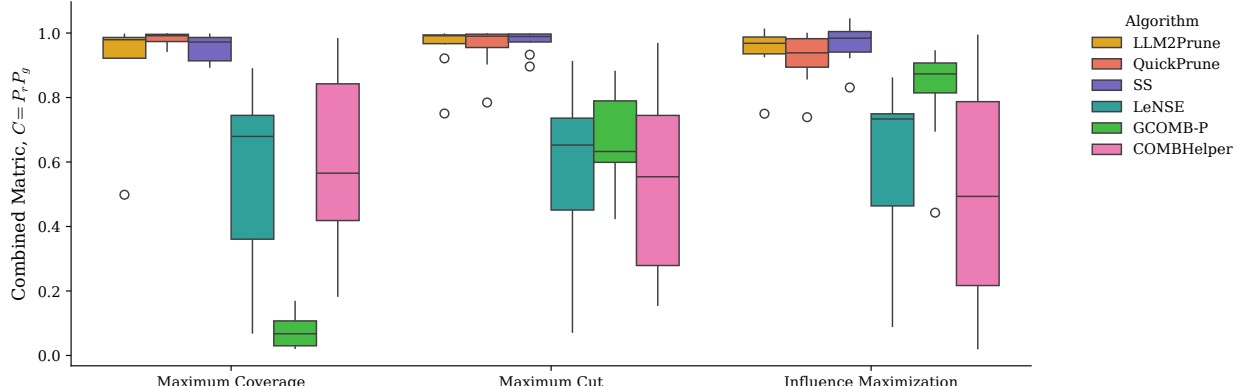

Figure 1: Comparison of pruning algorithms across three CO problems. The boxplots report the combined metric $C = P_r P_g$ (defined below), which balances solution quality and ground set reduction. LLM2Prune (ours) is competitive with classical approaches (QuickPrune, SS) and outperforms learning-based approaches (GCOMB, LeNSE, COMBHelper) by a large margin, without relying on domain knowledge.

The pruning problem seeks to construct a much smaller ground set $\mathcal{U}'$ with $|\mathcal{U}'| \ll |\mathcal{U}|$, such that solving the reduced instance yields nearly the same objective value:

$$\frac{f(\mathcal{OPT}(\mathcal{U}'))}{f(\mathcal{OPT}(\mathcal{U}))} = 1 - \epsilon$$

for a small approximation factor $\epsilon \geq 0$.

Existing pruning approaches are limited by scalability issues (Zhou et al., 2017; Nath & Kuhnle, 2025), reliance on handcrafted problem-specific features (Ireland & Montana, 2022; Tian et al., 2024), lack of problem-agnostic pipelines or weak performance on general graph optimization problems (Lauri & Dutta, 2019; Lauri et al., 2023; Manchanda et al., 2020), and poor adaptability to constraints. Thus, in this work, we are motivated by the following question:

*Is it possible to design a scalable pruning framework that avoids handcrafted features, delivers strong performance across general CO problems over graphs, and adapts easily to constraints?*

To address these limitations, we propose LLM2Prune, a framework that replaces prior supervised methods relying on problem-specific, hand-crafted features that limit generality across graph optimization problems and constraints In addition, it overcomes classical methods that use sequential pruning, which struggle with scalability. Our approach uses LLMs to automatically generate features for a downstream classifier, which is then applied to prune the ground set of a graph optimization problem. Given a problem definition, we employ an LLM-guided beam search over the feature space, directed by both performance evaluation and feature importance scores. These signals act as surrogate gradients, steering the search toward high-quality features that improve the performance of the downstream classifier. In contrast, prior works that employ LLMs for feature generation either rely on manually specified rules, derive new features only from existing ones, or use simple feedback loops that allow error propagation and lack feature-importance guidance. By comparison, our approach generates features in a guided search framework that balances exploration with structured signals, making it robust to noisy feedback and enabling consistent feature discovery.

**Contributions.** Our work makes the following contributions:

- We propose LLM2Prune, a framework that leverages the reasoning capabilities of LLMs to automatically generate features for a lightweight classifier that prunes the search space of CO problems over graphs. Unlike prior approaches, it avoids hand-crafted, problem-specific engineering and can easily adapt to constraints.

- We introduce an LLM-guided beam search strategy for feature generation that incorporates feature-importance score and performance evaluation. This approach overcomes the limitations of a simple feedback loop by maintaining diversity and stability in feature exploration.

- We evaluate LLM2PRUNE on diverse graph optimization problems under both size and knapsack constraints. Under size constraints, LLM2PRUNE achieves solution quality comparable to classical methods while enabling substantially faster pruning. It consistently outperforms state-of-the-art learning-based pruning approaches in solution quality and runtime. The ability of LLM2PRUNE to adapt naturally to additional constraints is demonstrated under knapsack constraints, where it achieves superior performance compared to existing methods.

**Organization.** The rest of this paper is organized as follows. In Section 2, we describe the relationship between our contribution and existing work. In Section 3, we present our algorithm, LLM2PRUNE, and conduct an empirical evaluation, comparing it to existing learned and classical heuristics for the pruning problem, followed by a discussion of limitations and future directions. Finally, in Section 5, we conclude the paper.

## 2 Related Work

Our work closely relates to two lines of research: pruning the search space for CO problems and automated feature engineering with LLMs. We emphasize that our approach is orthogonal to methods that directly generate algorithms using LLMs, which often exhibit degraded performance as graph size increases (Yang et al., 2023; Liu et al., 2024). Instead, we leverage the reasoning capabilities of LLMs to generate features for a learned algorithm for pruning.

**Automated Feature Engineering via LLMs.** Automated feature engineering generates useful features from raw data without human effort. Prior work with LLMs has focused primarily on tabular data, where new features are derived from existing ones. These approaches either rely on manually designed feature-generation rules (Zhang et al., 2023; 2024) or few-shot prompts (Han et al., 2024), and thus depend on hand-crafted design choices. Moreover, existing methods (Nam et al., 2024; Hollmann et al., 2023) suffer from error propagation caused by simple feedback loops.

In contrast, our method does not rely on existing features, a predefined search space, or few-shot examples. Instead, it uses feature-importance scores to identify useful features rather than depending solely on performance evaluation. To reduce error propagation and ensure stable feature generation, we incorporate beam search, which produces more robust and consistent results.

**Pruning Search Space for CO Problems.** To improve the scalability of heuristics, several works have proposed to prune the search space of CO problems and focus only on the reduced ground set. These approaches fall into two categories: classical and learning-based algorithms.

**Classical Approaches.** Zhou et al. (2017) propose a pruning algorithm based on submodularity to reduce the cost of submodular maximization under size constraints. Similarly, Nath & Kuhnle (2025) introduce a submodularity-based pruning algorithm that provides theoretical guarantees on both the fraction of the optimal value retained and the size of the resulting pruned ground set. The main limitation of these approaches is that they prune sequentially, which makes the process time-consuming. In contrast, our algorithm prunes in a single step. While this can produce a slightly larger pruned ground set in some cases, we demonstrate that our algorithm is orders of magnitude faster, particularly on large datasets.

**Learning-Based Approaches.** These algorithms can be divided into supervised learning methods and hybrid approaches that combine supervised and reinforcement learning. In the supervised setting, small instances are solved with exact or heuristic solvers, and the solutions are used to train a binary classifier that predicts whether a vertex or edge can be safely removed without compromising solution quality. These methods rely on hand-crafted, problem-specific features. For Maximum Clique Enumeration, Lauri & Dutta (2019) apply logistic regression with hand-crafted features to prune vertices, while Grassia et al. (2019); Lauri et al. (2023) extends this line of work with a multi-stage pruning strategy in which successive classifiers,

trained on graph-theoretic and statistical features, iteratively reduce the search space. Similarly, Fitzpatrick et al. (2021); Sun et al. (2021) train classifiers to prune edges unlikely to belong to the optimal tour in Traveling Salesman Problem instance.

The main limitation of these methods is their heavy reliance on problem-specific manual feature engineering and trial-and-error design, which restricts generalization. To address this, Tian et al. (2024) proposes COMBHELPER, a GNN-based framework with knowledge distillation and a problem-specific boosting module designed to generalize across CO problems over graphs. While effective, it still depends on computationally expensive Fourier Feature Mapping and problem-specific boosting.

Hybrid methods that integrate supervised and reinforcement learning are generally more problem-agnostic. For instance, GCOMB (Manchanda et al., 2020) uses a probabilistic greedy algorithm to generate training data, followed by a linear regression model that applies a weighted-degree heuristic to prune the set. A Q-learning network is then trained on this reduced search space to sequentially predict the solution. A key limitation is that the initial pruning depends on a handcrafted degree heuristic, which does not generalize well across CO problems (Ireland & Montana, 2022).

Similarly, LeNSE (Ireland & Montana, 2022) employs a two-stage learning process. First, a GNN is trained in a supervised manner to generate embeddings for subgraphs sampled from training graphs. Then, a policy network uses these embeddings to guide a local search procedure, moving from an initial subgraph to one more likely to contain the optimal solution. A major drawback, however, is its reliance on domain expertise to categorize subgraphs, set the number of samples per class, and tune critical hyperparameters such as subgraph size and embedding dimensionality separately for each dataset and problem.

As opposed to prior learning-based approaches, LLM2PRUNE makes the pruning process fully automated, requiring no domain knowledge, and can easily adapt to constraints.

## 3 LLM2Prune

The main goal of our work is to design a learning-based framework that prunes the search space through automated reasoning, making the pruning process generalize across graph optimization problems while accommodating additional constraints. We formalize the feature space search with LLMs as a decision tree, denoted by $\mathcal{T}$. Each node in the tree will have its own proposed feature set and classifier, along with a running log of everything tried so far, including good features, bad features, performance results, and importance scores, so the LLM can propose new features knowing both the current state and the full path of past attempts. Formally, each node is denoted by $n_{d,i}$, where $d \geq 0$ indicates the depth of the node in the tree and $i \in \{1, \ldots, N_d\}$ indexes the node among all $N_d$ nodes at depth $d$.

Each node $n_{d,i}$ is represented as a tuple

$$n_{d,i} = \left(\mathcal{F}_{d,i}, \mathcal{M}_{d,i}, \ g_{\theta_{d,i}}, \ \mathcal{H}_{d,i}\right),$$

where $\mathcal{F}_{d,i}$ is the proposed feature set, $g_{\theta_{d,i}}$ is the trained classifier on $\mathcal{F}_{d,i}$, and $\mathcal{M}_{d,i}$ denotes the performance metric of the classifier, defined as the combined metric $C = P_r \cdot P_g$ (see Section 4) evaluated on a held-out synthetic validation graph. The final component, $\mathcal{H}_{d,i}$, represents the history, defined as the accumulation of all feature sets, performance metrics, feature-importance vectors, and discarded features (if execution exceeds the timeout $\tau$) along the unique path from the root node $n_{0,1}$ to the node $n_{d,i}$:

$$\mathcal{H}_{d,i} = \bigcup_{\ell=0}^{d} \bigcup_{j \in \pi(\ell,i)} \left(\mathcal{F}_{\ell,j}, \ M_{\ell,j}, \ \mathrm{Imp}_{\ell,j}, \ \mathcal{F}_{\ell,j}^{\mathrm{fail}}\right),$$

where $\pi(\ell, i)$ denotes the ancestor index at depth $\ell$ along the path from the root to $n_{d,i}$, $M_{\ell,j}$ is the performance metric of the classifier, $\mathrm{Imp}_{\ell,j}$ is the feature-importance vector, and $\mathcal{F}_{\ell,j}^{\mathrm{fail}}$ is the set of features discarded at $n_{\ell,j}$ due to exceeding cut-off time.

The cumulative history $\mathcal{H}_{d,i}$ that records all past feature proposals, performance evaluations, and importance scores along the path from the root. This formulation ensures that the LLM generates new feature

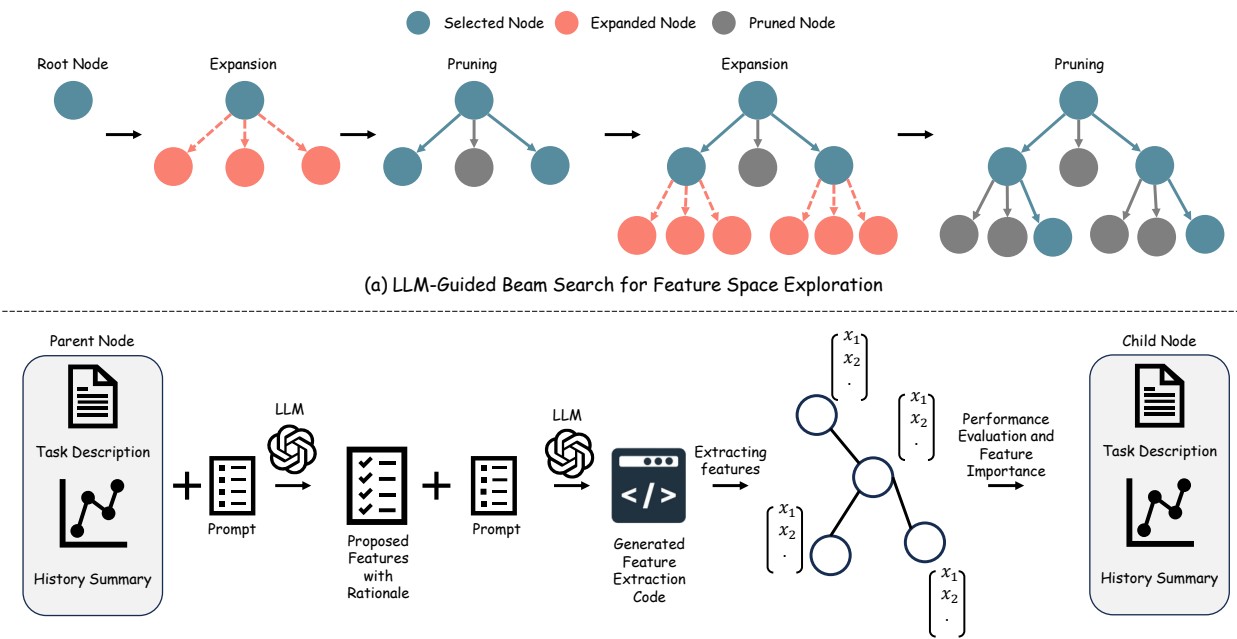

Figure 2: (a) Two phases of LL2Prune, illustrated with beam width $\beta = 2$ and expansion factor $\gamma = 3$. In the expansion stage, each node expands into $\gamma$ child nodes and in the pruning stage the beam grows by another depth, keeping its width size to $\beta$ after pruning out the candidate nodes with poor performances. (b) An overview of the expansion stage. Each parent node, consisting of a task description and a history summary, is used to prompt the LLM to propose new features. A second prompt generates executable feature extraction code, which is applied to produce features. These are evaluated for performance and importance, and the resulting history is carried forward to form child nodes.

proposals with full awareness of both the local information at the node and the trajectory that led to it. We condition on the full path rather than only the immediate parent because the LLM may otherwise re-propose features explored and discarded at earlier depths; providing the complete history of past attempts avoids this redundancy and reduces the number of repeated proposals during search.

Our algorithm begins at the root node and expands the search tree one depth level at a time in a breadth-first manner. At each depth $d$, the procedure consists of two phases: expansion and pruning (as shown in Figure 2A).

**Expansion.** The expansion phase of our algorithm follows the expansion step of standard beam search. Given an expansion factor $\gamma$, each node $n_{d,i}$ expands into $\gamma$ child nodes. These child nodes are produced directly by the LLM, which takes as input the task description with constraints and the accumulated history $\mathcal{H}_{d,i}$, together with a feature-generation meta-prompt $Z_{\text{feature}}$.

This meta-prompt does not simply request new features but provides higher-level guidance on how the LLM should generate new features. Specifically, $Z_{\text{feature}}$ (see Appendix A.1) instructs the LLM to enumerate and retain important features, remove redundant or uninformative ones, and propose new features derived from the current set of high-performing features. In addition, we include meta-instructions requiring the LLM to provide both a formal definition and a rationale for each proposed feature.

A key property of this step is that the LLM is stochastic: even under identical inputs, it can generate diverse outputs due to sampling mechanisms such as temperature, thereby broadening exploration of the feature space. Formally, the child feature sets are sampled as

$$\{\mathcal{F}_{d+1,j}\}_{j=1}^{\gamma} \sim \text{LLM}\big(Z_{\text{feature}}, \text{Task}, \text{Summary}(\mathcal{H}_{d,i})\big),$$

where $\text{Summary}(\mathcal{H}_{d,i})$ denotes the LLM-generated summary of the history at node $n_{d,i}$. We provide a meta-prompt $Z_{\text{summary}}$ (see Appendix A.2) that summarizes the history feedback into key points and insights. Instead of providing the entire feedback history from the root to a node, this approach keeps the feedback compact and makes the reasoning process easier for the LLM to follow.

Given a child node with its generated feature set $\mathcal{F}_{d+1,j}$, we prompt the LLM with meta-prompt $Z_{code}$ (see Appendix A.3) to produce executable code for each proposed feature $f \in \mathcal{F}_{d+1,j}$. Executing this code yields the feature values for the training data. To ensure computational efficiency, we impose a timeout threshold $\tau$. If execution exceeds $\tau$, the failure is recorded in the history $\mathcal{H}_{d+1,j}$ and the corresponding feature is discarded. This mechanism (i) keeps the pruning process lightweight by excluding costly features and (ii) prevents further exploration of features correlated with discarded ones. If the execution of a code block raises an error, this error is passed to the LLM for the next code generation iteration.

Traditional supervised approaches (Lauri et al., 2023; Tian et al., 2024) often assume that all nodes not included in the solution obtained from an exact or heuristic solver are equally unimportant, an assumption known to introduce label noise (Manchanda et al., 2020). While Manchanda et al. (2020) addresses this issue using a probabilistic greedy approach, where the loss of classifying each node is proportional to how frequently it is selected by solvers, we instead integrate semi-supervised learning to mitigate label noise. We then train a classifier $g_\theta$ on the features in $\mathcal{F}_{d+1,j}$. For each element $v \in \mathcal{U}$, the model outputs a probability distribution $p_\theta(y \mid v)$ over labels $y \in \{0,1\}$. After a warm-up period, we retain the high-confidence nodes

$$\mathcal{V}_{\text{conf}} = \{v \in \mathcal{V} : \max_y p_\theta(y \mid v) \geq \delta\},$$

where $\delta$ is a confidence threshold. For each $v \in \mathcal{V}_{\text{conf}}$, we assign a soft pseudo-label

$$\tilde{y}_v = p_\theta(y \mid v).$$

These soft pseudo-labels, together with the ground-truth labels on the supervised subset $\mathcal{V}_{\text{labeled}}$, guide further training and improve robustness by incorporating uncertainty rather than enforcing hard assignments.

The downstream classifier $g_\theta$ is implemented as a two-layer Graph Convolutional Network (GCN) (Kipf & Welling, 2016), operating on the LLM-proposed node-level features. For each node $v \in \mathcal{U}$, it outputs a probability $p_\theta(y = 1 \mid v)$ indicating how likely the node belongs in the pruned ground set.

We then validate the trained network and extract feature importance scores using an explainability algorithm, which provides insights into how different features influence the model's predictions. This information constitutes new feedback for the corresponding child node and is appended to its history. To extract feature-importance vectors, we apply GNNExplainer (Ying et al., 2019), which provides attribution scores indicating each feature's contribution to the classifier's predictions. These scores form the feedback signal $\text{Imp}_{d,i}$ used to guide subsequent feature generation.

**Pruning.** Let the set of nodes at depth $d$ be denoted by

$$\mathcal{N}_d = \{n_{d,1}, \ldots, n_{d,N_d}\}.$$

From this set, the algorithm maintains a beam

$$B_d \subseteq \mathcal{N}_d, \quad |B_d| = \beta,$$

where $\beta$ is the beam width. During pruning, the algorithm evaluates each candidate node $n_{d+1,j} \in \mathcal{C}_{d+1}$ using its performance metric $M_{d+1,j}$, and constructs the next beam $B_{d+1}$ by selecting top-$\beta$ nodes.

The process continues until the search reaches a maximum depth $D_{\max}$, which we treat as a hyperparameter. At this point, the algorithm terminates, and the best node overall is returned as

$$n^{\text{best}} = \arg \max_{n_{d,i} \in \mathcal{T}} M_{d,i}.$$

Finally, when beam search finishes, we keep only the best node found so far based on the performance metric. In Appendix F, we compare the beam-search variant against alternative configurations, e.g., a

---

**Algorithm 1** LLM2Prune

---

**Require:** Task description, ground set $\mathcal{U}$, beam width $\beta$, expansion factor $\gamma$, max depth $D_{\max}$, timeout $\tau$, confidence threshold $\delta$
**Ensure:** Pruned ground set $\mathcal{U}' \subseteq \mathcal{U}$
 1: Initialize root node $n_{0,1}$ with empty feature set and history $\mathcal{H}_{0,1} = \emptyset$
 2: $B_0 \leftarrow \{n_{0,1}\}$
 3: **for** $d = 0, 1, \ldots, D_{\max} - 1$ **do**
 4:     $\mathcal{C}_{d+1} \leftarrow \emptyset$
 5:     **for** each $n_{d,i} \in B_d$ **do**                                               ▷ Expansion
 6:         summary $\leftarrow$ LLM$(Z_{\text{summary}}, \mathcal{H}_{d,i})$
 7:         **for** $j = 1, \ldots, \gamma$ **do**
 8:             $\mathcal{F}_{d+1,j} \sim$ LLM$(Z_{\text{feature}}, \text{Task}, \text{summary})$
 9:             Generate and execute code for each $f \in \mathcal{F}_{d+1,j}$; discard features exceeding $\tau$
10:             Train $g_{\theta_{d+1,j}}$ on $\mathcal{F}_{d+1,j}$ with semi-supervised learning (threshold $\delta$)
11:             Compute $M_{d+1,j}$ and $\text{Imp}_{d+1,j}$; update $\mathcal{H}_{d+1,j}$
12:             $\mathcal{C}_{d+1} \leftarrow \mathcal{C}_{d+1} \cup \{n_{d+1,j}\}$
13:         **end for**
14:     **end for**
15:     $B_{d+1} \leftarrow$ top-$\beta$ nodes in $\mathcal{C}_{d+1}$ by $M$                       ▷ Beam pruning
16: **end for**
17: $n^{\text{best}} \leftarrow \arg\max_{n_{d,i} \in \mathcal{T}} M_{d,i}$
18: Fine-tune $g_{\theta^{\text{best}}}$ on the target training graph
19: **return** top-ranked nodes of $\mathcal{U}$ under $g_{\theta^{\text{best}}}$ as $\mathcal{U}'$

---

simple feedback loop and a no-feedback baseline, to isolate the effects of iterative feedback and feature-importance guidance. We emphasize that LLM2Prune is a general framework that can work with any classifier and any explainability algorithms applied to that classifier.

The beam search is performed once on synthetic graphs that match the structural distribution of the target domain. The metric $M$ serves as a selection criterion among candidate feature sets during search. Once the best feature set is identified, the classifier is fine-tuned on the actual target training graph and applied to prune the ground set. Algorithm 1 describes the full LLM2Prune pipeline.

## 4    Empirical Evaluation

In this section, we compare the empirical performance of LLM2Prune with existing methods for the pruning problem across three applications, following (Ireland & Montana, 2022; Manchanda et al., 2020; Nath & Kuhnle, 2025).

**Summary of Results.** For all experiments, we show that LLM2Prune typically prunes a large fraction of the ground set (often exceeding 90% on medium-to-large graphs, though pruning ratios vary by dataset, e.g., as low as ∼50% on the small Facebook graph under Maximum Coverage) while sacrificing very little objective value, achieving competitive performance or outperforming the competing methods. Under the size constraint, although classical approaches such as QUICKPRUNE (Nath & Kuhnle, 2025) and SUBMODULAR SPARSIFICATION (SS) (Zhou et al., 2017) outperform our method on the combined metric (defined below) in some cases, LLM2Prune can be orders of magnitude faster while remaining competitive, since classical approaches sequentially prune the ground set, as shown in Figure 3. Among learning-based pruning methods, LLM2Prune consistently outperforms GCOMB-P (Manchanda et al., 2020), LeNSE (Ireland & Montana, 2022), and COMBHELPER (Tian et al., 2024) across nearly all evaluated instances. Furthermore, under the knapsack constraint, LLM2Prune outperforms TOP-$k$ and QUICKPRUNE while retaining the flexibility and efficiency observed in the size-constrained setting, demonstrating that our framework generalizes naturally to different constraints.

Table 1: Comparison of pruning algorithms for size-constraint experiments. Highlighted are the results ranked **first**, **second** , and **third**. *Values are reported from Ireland & Montana (2022); Nath & Kuhnle (2025). "–" indicates that the corresponding algorithm obtained no reasonable result. Due to space constraints, we report the results for Maximum Cover in Appendix G, and the results are qualitatively similar.

| Dataset | Classical Approach | | | | | | Learning-Based Approach | | | | | | | | | | | |
|---|---|---|---|---|---|---|---|---|---|---|---|---|---|---|---|---|---|---|
| | QuickPrune | | | SS | | | LLM2Prune | | | GCOMB-P* | | | LeNSE* | | | COMBHelper* | | |
| | $P_r$ | $P_g$ | $C$ | $P_r$ | $P_g$ | $C$ | $P_r$ | $P_g$ | $C$ | $P_r$ | $P_g$ | $C$ | $P_r$ | $P_g$ | $C$ | $P_r$ | $P_g$ | $C$ |
| Maximum Cut | | | | | | | | | | | | | | | | | | |
| Facebook | 0.9886 | 0.7935 | 0.7845 | 0.9928 | 0.9027 | 0.8962 | 1.0004 | 0.7501 | 0.7504 | 0.8130 | 0.9500 | 0.7723 | 1.0000 | 0.0700 | 0.0700 | 1.0000 | 0.1538 | 0.1538 |
| DBLP | 1.0000 | 0.9954 | 0.9954 | 1.0000 | 0.9963 | 0.9963 | 0.9981 | 0.9841 | 0.9822 | 0.6460 | 0.9900 | 0.6395 | 0.9930 | 0.9200 | 0.9136 | 1.0000 | 0.1825 | 0.1825 |
| Deezer | 0.9996 | 0.9730 | 0.9726 | 0.9999 | 0.9855 | 0.9854 | 0.9999 | 0.9907 | 0.9906 | 0.8500 | 0.9900 | 0.8415 | 0.9750 | 0.7400 | 0.7215 | 1.0000 | 0.3754 | 0.3754 |
| Slashdot | 1.0000 | 0.9904 | 0.9904 | 1.0000 | 0.9889 | 0.9889 | 1.0000 | 0.9926 | 0.9926 | 0.6320 | 0.9900 | 0.6257 | 0.9900 | 0.6200 | 0.6138 | 0.7935 | 0.9929 | 0.7879 |
| Wiki | 0.9985 | 0.9037 | 0.9023 | 0.9981 | 0.9346 | 0.9328 | 1.0000 | 0.9214 | 0.9214 | 0.9200 | 0.9600 | 0.8832 | 0.9810 | 0.3900 | 0.3826 | 1.0000 | 0.7011 | 0.7011 |
| Twitter | 1.0000 | 0.9900 | 0.9900 | 1.0000 | 0.9893 | 0.9893 | 1.0000 | 0.9938 | 0.9938 | 0.6280 | 0.9900 | 0.6217 | 0.9870 | 0.4800 | 0.4738 | 1.0000 | 0.5542 | 0.5542 |
| YouTube | 1.0000 | 0.9994 | 0.9994 | 1.0000 | 0.9987 | 0.9987 | 1.0000 | 0.9991 | 0.9991 | 0.5360 | 0.9900 | 0.5306 | 0.9870 | 0.7900 | 0.7797 | 0.9990 | 0.9705 | 0.9695 |
| Skitter | 1.0000 | 0.9995 | 0.9995 | 1.0000 | 0.9991 | 0.9991 | 0.9966 | 0.9988 | 0.9954 | 0.4270 | 0.9900 | 0.4227 | 0.9740 | 0.7100 | 0.6915 | – | – | – |
| Influence Maximization | | | | | | | | | | | | | | | | | | |
| Facebook | 0.9919 | 0.7450 | 0.7390 | 0.9208 | 0.9027 | 0.8312 | 0.9995 | 0.7501 | 0.7498 | 0.9510 | 0.7300 | 0.6942 | 0.9790 | 0.0900 | 0.0881 | 1.0039 | 0.3248 | 0.3261 |
| DBLP | 0.8603 | 0.9946 | 0.8557 | 1.0495 | 0.9963 | 1.0456 | 0.9953 | 0.9984 | 0.9937 | 0.8630 | 0.9900 | 0.8544 | 0.9690 | 0.8900 | 0.8624 | 0.9873 | 0.0194 | 0.0192 |
| Deezer | 0.9384 | 0.9698 | 0.9101 | 1.0262 | 0.9855 | 1.0113 | 0.9759 | 0.9907 | 0.9668 | 0.8050 | 0.5500 | 0.4428 | 0.9720 | 0.7600 | 0.7387 | 0.9837 | 0.1093 | 0.1075 |
| Slashdot | 0.9984 | 0.9881 | 0.9865 | 0.9959 | 0.9889 | 0.9848 | 1.0215 | 0.9926 | 1.0139 | 0.9660 | 0.9800 | 0.9467 | 0.9660 | 0.7700 | 0.7438 | 1.0076 | 0.9875 | 0.9950 |
| Wiki | 1.0269 | 0.8831 | 0.9069 | 0.9863 | 0.9346 | 0.9218 | 1.0039 | 0.9214 | 0.9249 | 0.9690 | 0.9000 | 0.8721 | 0.9600 | 0.5100 | 0.4896 | 0.9969 | 0.6066 | 0.6047 |
| Twitter | 1.0045 | 0.9965 | 1.0010 | 0.9575 | 0.9893 | 0.9473 | 0.9817 | 0.9876 | 0.9695 | 0.9200 | 0.9800 | 0.9016 | 0.9660 | 0.4000 | 0.3864 | 0.9972 | 0.4947 | 0.4933 |
| YouTube | 0.9677 | 0.9994 | 0.9671 | 0.9846 | 0.9987 | 0.9833 | 1.0005 | 0.9991 | 0.9996 | 0.9330 | 0.9900 | 0.9237 | 0.9710 | 0.7500 | 0.7282 | 0.9992 | 0.9705 | 0.9697 |
| Skitter | 0.9813 | 0.9999 | 0.9812 | 1.0032 | 0.9991 | 1.0023 | 1.0035 | 0.9970 | 1.0006 | 0.8830 | 0.9900 | 0.8742 | 0.9830 | 0.7800 | 0.7667 | – | – | – |

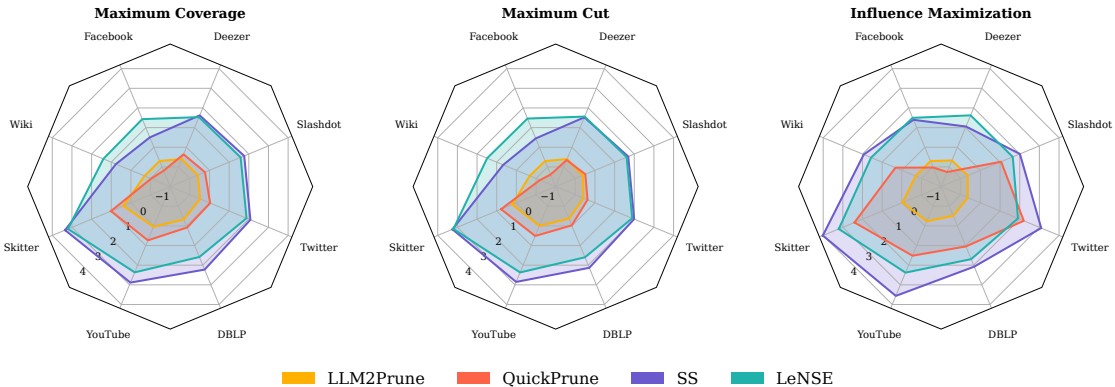

Figure 3: Runtime comparison (log scale) of pruning algorithms on Maximum Coverage, Maximum Cut, and Influence Maximization across eight datasets.

**Evaluation Metrics.** Following Ireland & Montana (2022); Nath & Kuhnle (2025), we evaluate the algorithms using three performance metrics, where higher values indicate better performance. The first is the pruning approximation ratio $P_r$, defined as the ratio of the objective value obtained from the pruned ground set $\mathcal{U}'$ to that from the original ground set $\mathcal{U}$, both computed by a heuristic $\mathcal{H}$, i.e., $P_r = f(\mathcal{H}(\mathcal{U}'))/f(\mathcal{H}(\mathcal{U}))$. The second is the pruned fraction $P_g$, which measures the proportion of the original ground set that is removed. Finally, to avoid trivial solutions that maximize one metric while disregarding the other (e.g., pruning nothing or pruning everything), we use the combined metric $C = P_r P_g$. We want to highlight that $C$ can exceed one than one in cases where the heuristic can find a better solution on the pruned ground set than on the full ground set. This is possible because the heuristic does not guarantee an exact solution, and with less noise, it may find a better solution on the reduced set. Similar findings are reported in Ireland & Montana (2022).

**Applications.** We evaluate our algorithm on three problems: Maximum Coverage (MaxCover), Maximum Cut (MaxCut), and Influence Maximization (IM), under both size and knapsack constraints, following Manchanda et al. (2020); Ireland & Montana (2022). Detailed specifications of these applications are provided in Appendix B, along with the optimal features identified by our framework.

**Discovered Features.** A key strength of LLM2PRUNE is that it generates features that are not generic graph statistics but are grounded in the problem objective and constraints. For MaxCut under the size constraint, it discovers *degree* and *random cut expectation*: these two features are complementary rather than redundant, as degree captures raw connectivity while random cut expectation measures the expected marginal contribution to the cut value under a random partition, integrating both topology and the cut objective in a way that degree alone cannot. For IM under the size constraint, it identifies *degree* and *average incoming activation probability*: degree reflects structural reach, while incoming activation probability encodes cascade dynamics (a node's susceptibility to being activated by its neighbors), representing orthogonal information about graph topology and the propagation model respectively. Under the knapsack constraint, LLM2PRUNE automatically encodes the cost-benefit trade-off: for MaxCov it selects *degree* and *coverage per cost* (capturing raw reach and cost-efficiency as distinct quantities), and for IM it selects *degree-to-weight ratio*, *outgoing activation probability sum*, and *in-out degree difference*, which encode cost-efficiency, influence-spread capacity, and directional propagation asymmetry, three distinct aspects of the knapsack IM problem. The feature-generation meta-prompt explicitly instructs the LLM to remove redundant features and ensure all proposals are distinct (see Appendix A.1, rules 2 and 5), which promotes complementarity in the selected sets. That said, minimality is not formally guaranteed: in higher-dimensional cases such as MaxCut under the knapsack constraint, where six features are selected (including both *degree* and *normalized degree*), some partial overlap may remain. Full feature lists are provided in Appendix B.

**Baselines and Prior Methods.** We compare our algorithm with methods that accelerate heuristics by pruning the ground set rather than directly optimizing the objective function. Our evaluation includes the following classical and learning-based approaches: QUICKPRUNE, SUBMODULAR SPARSIFICATION (SS), GCOMB-P (Manchanda et al., 2020), LENSE (Ireland & Montana, 2022), and COMBHELPER (Tian et al., 2024). Details of each baseline are provided in Appendix C. For the general knapsack constraint, prior methods do not generalize, except for QUICKPRUNE, so we compare our algorithm only with QUICKPRUNE.

**Heuristics.** For both size and knapsack constraints, we follow (Nath & Kuhnle, 2025): for MaxCover and MaxCut, we use STANDARD GREEDY (Nemhauser et al., 1978); for IM, we adopt IMM (Tang et al., 2015); and under the knapsack constraint, we employ Khuller et al. (1999) for IM and MaxCover, and Pham et al. (2023) for MaxCut.

**Datasets.** We evaluate our approach using real-world datasets from the Stanford Large Network Dataset Collection (Leskovec & Sosič, 2016) for the CO problems on graphs, following the experimental setup of Ireland & Montana (2022). A summary of all datasets used can be found in Appendix D.

**Common Setup.** In all experiments, our algorithm and the baselines are trained to create a pruned ground set for CO problems with a budget of $k = 100$, following Ireland & Montana (2022). The downstream classifier, a two-layer GCN (Kipf & Welling, 2016), operates on the LLM-proposed node-level features. To extract feature-importance vectors, we employ GNNExplainer (Kipf & Welling, 2016). For each dataset, we adopt the train–test splits from Ireland & Montana (2022). We experiment with two LLMs, GPT-5-nano and LLaMA-3.3-70B-Instruct, to propose features, generate code, and summarize feedback. We report results using GPT-5-nano in the main paper. The two models achieve effectively comparable performance (as shown in Figure 6). Results in Tables 1–3 report single-run point estimates; variance across independent runs is characterized for the beam search in Appendix F (Figure 5).

Since running beam search for features on each instance would be costly, and our main motivation is to reduce computation, we instead perform feature-space search for each problem using the Holme–Kim random graph model (Holme & Kim, 2002). We select this model because it closely resembles large-scale social networks: it exhibits both the scale-free degree distribution and the high clustering coefficient that are empirically observed in real-world social graphs (Holme & Kim, 2002). After identifying the feature set, we fine-tune the downstream classifier for each instance. Appendix E provides details on the network architecture, training and data generation hyperparameters, LLM settings, beam search, and baseline configurations.

### 4.1 Evaluation on Size Constraint

In this subsection, we first present our experiments under the size-constraint setting. Under this setting, each element in the ground set has a unit cost. All our baselines except QUICKPRUNE, which works for both size and knapsack constraints, are designed specifically for the size constraint.

**Result Analysis.** From Table 1, we observe that LLM2PRUNE performs competitively with classical methods such as QUICKPRUNE and SS. However, both approaches rely on sequential submodularity-based pruning, which is orders of magnitude slower than our method on large datasets such as YouTube and Skitter (as shown in Figure 3). In contrast, LLM2PRUNE achieves comparable or better performance without relying on hand-crafted features, demonstrating that learning-based pruning can be both effective and efficient.

Within the learning-based category, LLM2PRUNE outperforms prior methods in both combined metric and runtime. GCOMB-P and COMBHELPER frequently underperform, while LENSE is more stable but generally results in lower combined-metric values. Unlike other learning-based methods that depend on handcrafted features, LLM2PRUNE balances efficiency with strong performance, effectively narrowing the gap with classical methods.

**Runtime Analysis.** Figure 3 reports the runtime of the four strongest algorithms across all applications. Classical methods are generally slower: SS is consistently the slowest, while QUICKPRUNE is faster but scales poorly on large graphs. Among learning-based methods, LLM2PRUNE is orders of magnitude faster than LENSE while maintaining higher solution quality, striking a strong balance between scalability and accuracy. We note that GCOMB-P has faster raw inference time on small graphs (e.g., $0.002\,\mathrm{s}$ vs. $0.194\,\mathrm{s}$ on Facebook; see Table 5); however, LLM2PRUNE consistently achieves a higher combined metric across the full benchmark. Compared to other learning-based baselines, it also achieves the most consistent performance across applications. We report the absolute runtime analysis, including an inference-time breakdown by component (see Appendix G.3).

**Multi-budget Scores.** Following Ireland & Montana (2022), we evaluate whether heuristics can also achieve close-to-optimal results for budgets lower than the one used to train the classifier. In our experiments, the classifier is trained to identify a pruned ground set for a budget of 100. We then test whether the pruned ground set returned by the classifier can also be used for budgets below 100. We observe that the ground set provided by LLM2PRUNE remains optimal for budgets lower than the training budget in nearly all instances. We report the multi-budget scores (see Appendix G.5). The results are qualitatively similar to our experiments under the knapsack constraint.

### 4.2 Evaluation on Knapsack Constraint

For the knapsack constraint, we adopt the cost model introduced by Nath & Kuhnle (2025); Yaroslavtsev et al. (2020). Let $G = (V, E)$ be a graph with vertex set $V$ and edge set $E$. For each node $v \in V$, let $N(v) = \{u \in V : (u, v) \in E\}$ denote its neighborhood. We define the cost of node $v$ as $c(v) = \frac{\beta}{|V|}\big(|N(v)| - \alpha\big)$, where $\alpha = \frac{1}{20}$ (i.e., $\alpha = 0.05$) is a fixed offset parameter, and $\beta > 0$ is a normalization factor chosen such that $c(v) \geq 1$ for all $v \in V$. This cost model is more realistic for real-world graphs, where some nodes are inherently more expensive, for example, hubs in social networks or highly connected routers in communication networks. In contrast, a uniform cost unrealistically treats them as cheaply as sparsely connected nodes, which biases solutions toward hubs and can trivialize the optimization.

**Result Analysis.** As stated above, only QUICKPRUNE can be adopted to the knapsack constraint, so we compare our algorithm with QUICKPRUNE. We also compare it with another heuristic, TOP-k, following Nath & Kuhnle (2025).

Figure 4 shows that LLM2PRUNE outperforms TOP-k across all applications, and outperforms QUICKPRUNE for MaxCov and IM, while slightly lagging behind in MaxCut. We highlight that, unlike other learning-based methods, LLM2PRUNE can easily adapt to the knapsack constraint by simply modifying the task description provided to the model.

**Limitations and Future Directions.** A main limitation of LLM2PRUNE is the reliance on beam search, which is more computationally expensive than a simple feedback loop. As shown in Appendix F, simple

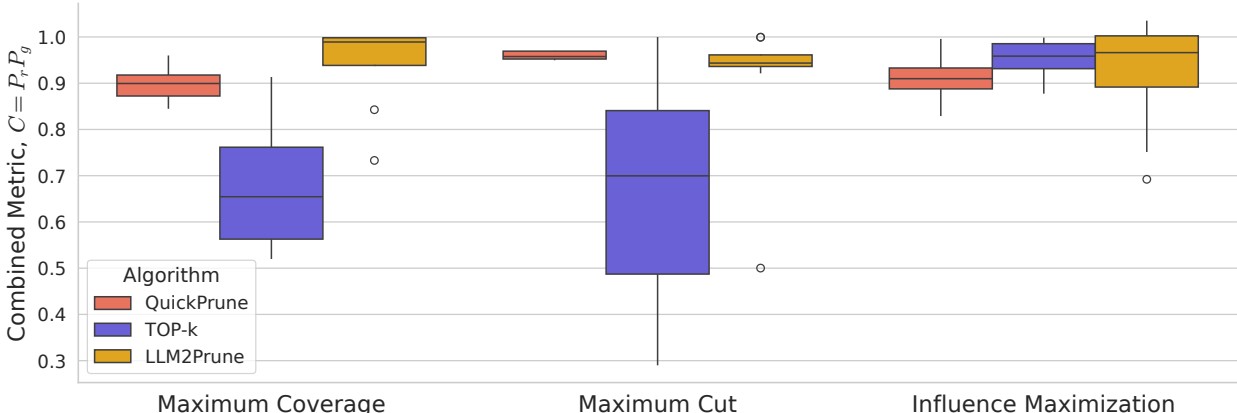

Figure 4: Combined metric for LLM2Prune, QuickPrune, and TOP-k under the knapsack constraint across MaxCov, MaxCut, and IM. LLM2Prune consistently outperforms TOP-k, matches or slightly exceeds QuickPrune on MaxCov and IM, and is comparable to QuickPrune on MaxCut.

feedback loops often fail to recover when early iterations contain reasoning errors, whereas beam search mitigates this issue by maintaining multiple candidate paths, at the cost of additional runtime. We note, however, that this evidence is indirect: our ablation compares beam search against a feedback loop but does not include a controlled experiment in which early branches are intentionally degraded. The resilience of beam search to unlucky initializations therefore remains empirically supported but not directly verified through perturbation. An important future direction is to reduce this overhead by leveraging reasoning tokens or similar mechanisms to improve the reliability of single-path feedback loops. Such approaches could capture richer intermediate reasoning without requiring multiple branches, thereby achieving the stability of beam search with significantly lower computational cost. Finally, the quality of the discovered features depends on the capability of the underlying LLM. If the model produces repetitive, hallucinated, or syntactically invalid feature proposals, the beam search may converge to a suboptimal feature set. While results are consistent across GPT-5-nano and LLaMA-3.3-70B-Instruct (Figure 6), weaker or smaller models may yield less reliable feature generation, and we have not systematically evaluated this degradation curve. Beyond computational cost, LLM2Prune is also sensitive to the choice of synthetic proxy used during feature discovery. When this proxy is structurally mismatched to the target graph family (for instance, Erdős–Rényi graphs, which have homogeneous degree distributions and lack clustering or community structure), the discovered features do not transfer well to real-world graphs. LLM2Prune therefore generalizes within a structural family but degrades under cross-family distribution shift. Furthermore, we have not systematically explored adversarial or structurally pathological cases within the intended regime (i.e., graphs that share the scale-free and clustered structure of HK graphs). It therefore remains an open question whether such graphs exist for which pruning fails sharply despite nominal structural similarity to the training proxy.

## 5 Conclusion

We present LLM2Prune, a general framework for accelerating heuristics on graph optimization problems without problem-specific domain knowledge. We prompt the LLM to propose candidate feature sets, and, guided by performance evaluation and feature importance from an explainer algorithm, a downstream classifier learns to identify nodes that can be safely pruned without degrading solution quality. Standard heuristics are then applied to the reduced search space to obtain the final solution. Across real-world datasets for diverse CO problems over graphs under both size and knapsack constraints, LLM2Prune consistently outperforms prior learning-based methods and remains competitive with classical approaches. Moreover, the framework adapts easily to new constraints, unlike other learning-based algorithms, and can even surpass classical methods tailored to those constraints. In addition, LLM2Prune achieves substantial runtime reductions on large-scale graphs. It preserves near-optimal solution quality even when pruning over 90% of the ground set. These results highlight both the efficiency and reliability of the approach.

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

# A Meta-prompts

## A.1 Feature Generation Prompt

In Listing 1, we present the feature-generation meta-prompt $Z_{\text{feature}}$, which instructs the LLM to generate a feature set based on the provided feedback.

```
f'''You are an expert in graph neural networks and combinatorial optimization.

Consider the {problem}, defined as ({problem_definition}).
Propose node-level features for a GNN binary classifier that predicts nodes
likely to be in the optimal solution.

The heuristic can only select nodes from the reduced candidate set provided
by the GNN. The goal is to shrink the candidate set while ensuring that the
heuristic still achieves the same objective value.

Feedback from previous iterations:{explainer_feedback} if {explainer_feedback} else ""
Refinement rules:
1) Keep high-importance features.
2) Avoid proposing features that are too similar to existing or previously failed ones.
3) Replace or remove low-importance features.
4) Add new features inspired by important patterns.
5) Ensure all features are distinct and non-redundant."

Return the output as a JSON array of objects with keys:
- "feature": feature name in snake_case
- "definition": a one-line definition of how to compute it
- "reason": 1-2 sentences explaining why it is useful

Do not include any text outside the JSON array.
'''
```

Listing 1: Prompt for generating node-level features for a GNN binary classifier.

## A.2 Summary Generation Prompt

In Listing 2, we present the meta-prompt $Z_{\text{summary}}$, which summarizes the feedback history.

```
f'''You are an expert in graph neural networks and combinatorial optimization.

Summarize the following feedback from previous iterations:
{cumulative_feedback}

Provide a concise summary of the key points and insights.
'''
```

Listing 2: Prompt for summarizing feedback from previous iterations.

## A.3 Code Generation Prompt

Listing 3 presents the meta-prompt $Z_{code}$, which instructs the LLM to generate efficient feature-extraction code from a given feature definition.

```
f'''The input is a weighted NetworkX graph G where each node has an attribute weight and an integer variable
↪  budget is provided.

Feature name: {feature}
Feature definition: {definition}

"Write Python code for a function extract_feature(G) that computes this feature for ALL nodes in G. The
↪  function must return a NumPy array with one value per node, ordered to align with the iteration order of
↪  G.nodes()."
"Keep the implementation efficient; avoid expensive computations."
"DO NOT INCLUDE ANY EXPLANATIONS OR COMMENTS."
'''
```

Listing 3: Prompt for generating feature-extraction code.

## B  Applications

In this section, we formally define the three applications discussed in the paper. A combinatorial optimization problem on a graph can be represented by an undirected graph $G = (V, E)$, where $V$ denotes the set of vertices, $E$ the set of edges, and each node $v \in V$ is associated with a cost $c(v)$.

**Maximum Coverage (MaxCov).** Given a budget $k$, the objective is to select a subset of nodes $S \subseteq V$ that maximizes

$$f(S) = |\{v \mid v \in S \vee \exists (u, v) \in E, u \in S\}|,$$

subject to the budget constraint $\sum_{v \in S} c(v) \leq k$.

Under the size constraint, LLM2PRUNE identifies closed neighborhood size, which includes both the node itself and its immediate neighbors and thus reflects local coverage potential, as the optimal feature. Under the knapsack constraint, the selected features are degree and coverage per cost.

**Maximum Cut (MaxCut).** Given a budget $k$, the goal is to find a subset of nodes $S \subseteq V$ that maximizes

$$f(S) = |\{(u, v) \in E \mid v \in S, u \in V \setminus S\}|,$$

while satisfying $\sum_{v \in S} c(v) \leq k$.

Under the size constraint, LLM2PRUNE identifies degree and random cut expectation as the optimal feature set, where random cut expectation measures the expected contribution of a node to the cut value under a randomly generated partition, capturing its average marginal influence. Under the knapsack constraint, the optimal features are degree, weight, degree to weight ratio, normalized degree, neighbor weight sum, and clustering coefficient.

**Influence Maximization (IM).** Given a budget $k$, edge propagation probabilities $p(u, v)$ for each $(u, v) \in E$, and a cascade model $\mathcal{C}$, the goal is to select a subset $S \subseteq V$ that maximizes the expected influence spread,

$$f(S) = \mathbf{E}[\sigma(S)],$$

where $\sigma(S)$ denotes the number of nodes influenced by $S$, and $\sum_{v \in S} c(v) \leq k$. In our experiments, we adopt the independent cascade model (Kempe et al., 2003), where the edge-level propagation probabilities assigned following the approaches of Ireland & Montana (2022); Manchanda et al. (2020).

Under the size constraint, LLM2PRUNE identifies degree and average incoming activation probability as the optimal features, where average incoming activation probability denotes the mean probability that a node becomes activated by its neighbors, capturing its susceptibility to influence. Under the knapsack constraint, the selected features are degree to weight ratio, outgoing activation probability sum, and in–out degree difference.

## C   Baselines

In this section, we provide detailed descriptions of each algorithm discussed in the paper.

**GCOMB-P.** For each training graph $G_i = (V, E)$ and a given budget $b$, we begin by ranking all vertices in descending order according to their total outgoing edge weight (or degree, in the case of unweighted graphs). The rank of a vertex $v$, denoted rank($v$), indicates its position in this ordered list, where a lower rank value corresponds to a more influential or connected node.

Next, we employ a stochastic solver $m$ times to generate $m$ different solution sets $\{S^{(1)}, S^{(2)}, \ldots, S^{(m)}\}$ for the same budget $b$. Since the solver introduces randomness, each run may produce a slightly different subset of nodes. Our goal is to identify all nodes that could possibly appear in any of these solutions.

To capture this, we define

$$r_b^{G_i} = \max_{v \in \cup_j S^{(j)}} \text{rank}(v),$$

which represents the highest-ranked (i.e., least promising) vertex among all nodes selected by the solver across the $m$ runs for graph $G_i$. Intuitively, $r_b^{G_i}$ marks the cutoff rank beyond which nodes are unlikely to be part of any optimal or near-optimal solution.

We repeat this computation for every training graph $G_i \in G_{\text{train}}$ and define

$$r_b = \max_{G_i \in G_{\text{train}}} r_b^{G_i},$$

ensuring that $r_b$ generalizes across the entire training distribution for the given budget $b$.

To extend this relationship across different budgets, we compute $r_b$ for a sequence of budgets and record the corresponding pairs $(b, r_b)$. Both the budgets and ranks are then normalized by the total number of nodes in each graph, allowing the learned relationship to generalize across graphs of varying sizes.

During inference, when an unseen budget $b$ is provided, we use linear interpolation over the known $(b, r_b)$ pairs to estimate the corresponding cutoff rank $r_b$. This interpolated value determines the subset of top-ranked nodes to consider as potential solution candidates for the new instance.

**LeNSE.** Ireland & Montana (2022) proposed a framework for learning to extract a small portion of the original graph that is most likely to contain the optimal solution, enabling efficient downstream optimization using any heuristic. The method first constructs a supervised learning stage followed by a reinforcement learning (RL) stage.

In the first stage, an encoder network is trained to classify subgraphs according to their quality relative to the optimal solution. To build this training data, the optimal solutions are first obtained for all graphs in the training distribution. Then, multiple random subgraphs are generated by sampling different fractions of nodes that overlap with the optimal solution, where each subgraph includes the sampled nodes and their 1-hop neighbors. Each subgraph is assigned to a class based on the ratio between its objective value and that of the original full-graph solution (e.g., high-, medium-, or low-quality). The encoder is trained using these labeled subgraphs so that its embeddings capture structural and functional information correlated with solution quality.

In the second stage, the learned encoder is used to provide the embedding space in which a reinforcement learning agent operates. The RL agent starts from a subgraph induced by a fixed number of randomly selected nodes and their 1-hop neighbors. At each step, the agent updates this subgraph by replacing one vertex with one of its neighbors, thereby navigating the space of subgraphs. The goal of the agent is to discover a subgraph whose embedding is close to those of high-quality (i.e., near-optimal) subgraphs identified during the supervised training phase. In this way, the encoder provides a learned representation of subgraph quality, while the RL agent exploits this representation to efficiently search for promising regions of the graph likely to contain the optimal solution.

**QuickPrune.** Nath & Kuhnle (2025) proposed a submodularity-based pruning algorithm that filters elements based on their marginal contribution relative to cost, ensuring that uninformative or redundant

elements are discarded. To prevent the selected set from growing excessively, the algorithm periodically deletes weaker elements when the accumulated gain exceeds a threshold governed by a deletion parameter.

**Submodular Sparsification (SS).** Zhou et al. (2017) introduced a randomized pruning framework for large-scale submodular maximization based on a novel structure called the submodularity graph. The submodularity graph is a weighted directed graph $G(V, E, w)$ constructed from a normalized submodular function $f : 2^V \to \mathbb{R}_+$, where each node in $V$ represents an element of the ground set, and each directed edge $(u, v) \in E$ is assigned a weight $w(u, v) = f(v \mid u) - f(u \mid V \setminus u)$. Since computing all pairwise weights requires quadratic time, the authors proposed a randomized approximation scheme. In each iteration, a random subset of nodes is sampled and temporarily removed from the ground set, while the least influential elements, identified based on their relationships in the sampled subgraph, are discarded. The process continues until the ground set size falls below a predefined threshold, at which point the remaining elements are merged with the pruned subset to form the reduced ground set.

**COMBHelper.** Tian et al. (2024) proposed a learning-based framework that trains a GNN for vertex classification in combinatorial optimization problems, where the goal is to predict nodes that are likely to be part of the optimal solution. To improve scalability, the authors applied knowledge distillation to transfer knowledge from a larger model to a smaller GNN and incorporated problem-specific weight boosting to further enhance performance.

# D   Datasets

Table 2 summarizes the graph datasets used in our empirical evaluation, including the sizes of their respective training and testing splits for traditional combinatorial optimization problems on graphs. Following Ireland & Montana (2022), we determine the proportion of edges from each original graph allocated to the training and testing sets.

Table 2: The real-world graphs used to perform our experiments.

| Graph | Train Size | | Test Size | |
| --- | --- | --- | --- | --- |
| | Vertices | Edges | Vertices | Edges |
| Facebook | 3847 | 26470(30%) | 4002 | 61764 |
| Wiki | 4891 | 30228(30%) | 6358 | 70534 |
| Deezer | 48870 | 149460(30%) | 53511 | 348742 |
| SlashDot | 47546 | 140566(30%) | 67640 | 327988 |
| Twitter | 55827 | 134229(10%) | 80712 | 1208067 |
| DBLP | 63004 | 41994(10%) | 315305 | 1007872 |
| YouTube | 185193 | 179257(06%) | 1098104 | 2808367 |
| Skitter | 147604 | 110952(01%) | 1694318 | 10984346 |

# E   Network Architectures And Hyper-parameters

**Beam Search.** Beam search hyperparameters are determined using a grid search over $\beta$, $\gamma$, and $D$ for Maximum Coverage problem. We select $\beta = 3$, $\gamma = 2$, and $D = 5$ based on their stable performance across datasets and constraints, avoiding over-specialization to any single problem. This configuration is then fixed across all reported experiments.

**Training Data Generation.** For feature space search, we use the Holme-Kim random graph model with $n = 10,000$ nodes and hyperparameters $m = 4$ and $p = 0.01$ for training and validation.

**LLM Settings.** We run with GPT-5-nano and LLaMA-3.3-70B-Instruct with default parameters provided in the documentation.

**GNN.** The architecture consists of two layers, each containing a graph convolutional layer with ReLU activation and 16 hidden channels. The network is trained using the Adam optimizer with learning rate

$10^{-3}$ and weight decay $5 \times 10^{-4}$. During beam search, the GNN is trained for 1,000 epochs on the synthetic Holme–Kim graph. During per-dataset fine-tuning on the target training graph, training runs for 10,000 epochs.

**Timeout Mechanism.** We set the cutoff time as 5 seconds to bound feature-space exploration and ensure practical runtime.

**Semi-supervised Learning.** During training, we employ pseudo-labeling with a confidence threshold of $\delta = 0.90$ and select the top $k = 100$ most confident predictions at each iteration after a warm-up period of 500 epochs.

**Computational Environment.** All GNN training and inference experiments were run on a single NVIDIA A6000 GPU (48 GB VRAM). Beam search LLM calls were made via API. All methods were evaluated under identical resource constraints (same number of CPU threads and GPU), with the sole exception of SS, which requires additional threads on large graphs as noted in Table 5. .

**Testing Settings.** We adopt a progressive expansion strategy to determine an effective candidate ground set for heuristic evaluation. Starting with the top 500 nodes ranked by their predicted probabilities, the heuristic is first applied to obtain an initial objective value. The candidate set is then progressively expanded to include the top $1,000$, $2,000$, and $5,000$ nodes, respectively. After each expansion, the heuristic is re-evaluated, and the relative improvement in the objective value is measured. The process continues until the improvement falls below 1%, at which point the current subset is selected as the final ground set. This procedure ensures a balance between computational efficiency and solution quality.

# F   Ablation Study

We conduct an ablation study to analyze the contribution of the major components of LLM2Prune. Specifically, we evaluate the following variants of our framework using GPT-5-nano.

- **Single-shot prompt:** A single LLM query is used to generate features without any iterative refinement or feedback.

- **Simple feedback loop with performance feedback:** The LLM iteratively refines features based solely on downstream performance metrics from the learned pruning model.

- **Simple feedback loop with performance feedback and feature importance:** In addition to performance metrics, feature-importance scores from the classifier are provided to guide feature refinement.

- **Beam search:** Multiple candidate feature sets are maintained at each iteration, with beam expansion guided by performance and feature-importance feedback.

To ensure a fair comparison, all variants use the same total number of API calls. Concretely, beam search with $\beta = 3$, $\gamma = 2$, $D = 5$ makes $\beta \cdot \gamma \cdot D = 30$ feature-generation calls, 30 code-generation calls, and $\beta \cdot D = 15$ summary calls per run; the simple feedback loop and single-shot baselines receive the same total call budget distributed as a flat sequence of refinement iterations. Each of the five runs corresponds to a full re-execution of the entire pipeline, including LLM-driven feature search (with new LLM calls and new feature proposals) and downstream GNN training, so the reported variance captures both LLM stochasticity and training variability jointly. In Figure 5, we report the combined metric across three CO problems under knapsack constraints. The results show that beam search consistently achieves higher and more stable performance, with median values close to 1.0 and minimal variance across runs, indicating that the pipeline is stable with respect to the stochasticity of LLM-driven feature generation.

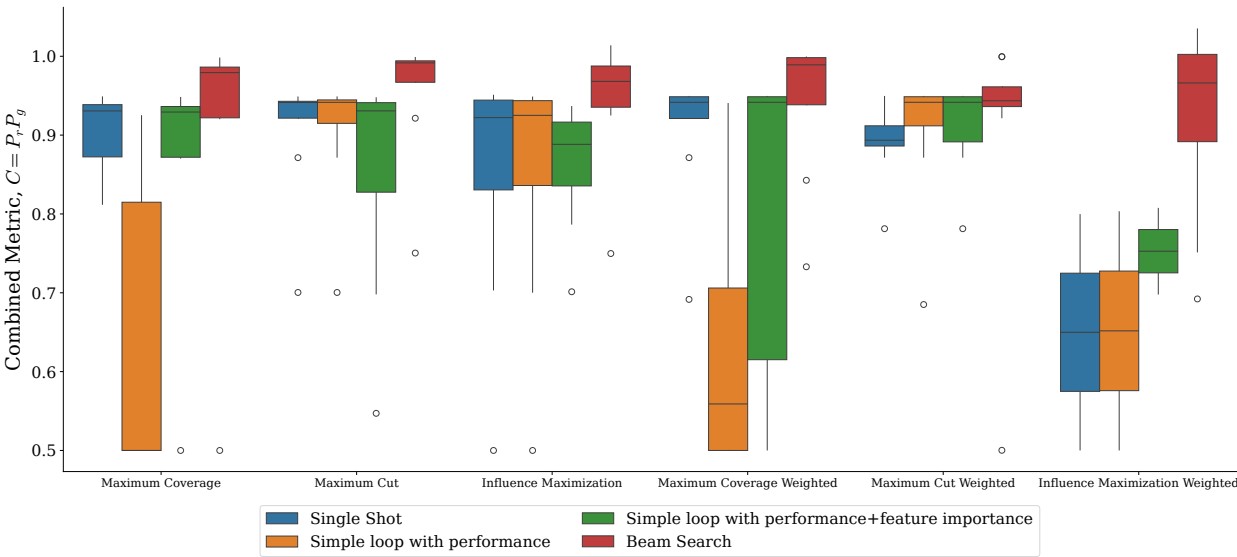

Figure 5: Performance comparison of different variants of LLM2PRUNE.

# G  Additional Tables and Plots

## G.1  Evaluation on Size Constraint

In Table 3, we represent the performance of the pruning algorithms on Maximum Coverage under size constraint.

Table 3: Comparison of pruning algorithms for size-constraint experiments. Highlighted are the results ranked **first**, **second** , and **third**. *Values are reported from Ireland & Montana (2022); Nath & Kuhnle (2025). "–" indicates that the corresponding algorithm obtained no reasonable result.

| | Classical Approach | | | | | | Learning-Based Approach | | | | | | | | | | | |
|---|---|---|---|---|---|---|---|---|---|---|---|---|---|---|---|---|---|---|
| Dataset | QuickPrune | | | SS | | | LLM2Prune | | | GCOMB-P* | | | LeNSE* | | | COMBHelper* | | |
| | $P_r$ | $P_g$ | $C$ | $P_r$ | $P_g$ | $C$ | $P_r$ | $P_g$ | $C$ | $P_r$ | $P_g$ | $C$ | $P_r$ | $P_g$ | $C$ | $P_r$ | $P_g$ | $C$ |
| | | | | | | | Maximum Coverage | | | | | | | | | | | |
| Facebook | 1.0000 | 0.9953 | 0.9953 | 0.9884 | 0.9027 | 0.8922 | 0.9966 | 0.5002 | 0.4985 | 0.9270 | 0.0700 | 0.0649 | 0.9660 | 0.0700 | 0.0676 | 1.0000 | 0.3840 | 0.3840 |
| DBLP | 0.9951 | 0.9957 | 0.9908 | 0.9945 | 0.9963 | 0.9908 | 1.0000 | 0.9841 | 0.9841 | 0.9990 | 0.0300 | 0.0300 | 0.9900 | 0.9000 | 0.8910 | 1.0000 | 0.1818 | 0.1818 |
| Deezer | 0.9606 | 0.9797 | 0.9411 | 0.9870 | 0.9855 | 0.9727 | 0.9930 | 0.9813 | 0.9745 | 0.9940 | 0.1300 | 0.1292 | 0.9790 | 0.7500 | 0.7343 | 1.0000 | 0.7278 | 0.7278 |
| Slashdot | 1.0000 | 0.9925 | 0.9925 | 0.9824 | 0.9889 | 0.9715 | 0.9914 | 0.9926 | 0.9840 | 1.0000 | 0.0200 | 0.0200 | 0.9790 | 0.6900 | 0.6755 | 1.0000 | 0.9844 | 0.9844 |
| Wiki | 0.9998 | 0.9422 | 0.9420 | 0.9559 | 0.9346 | 0.8934 | 0.9988 | 0.9214 | 0.9202 | 0.9900 | 0.0300 | 0.0297 | 1.0940 | 0.3400 | 0.3720 | 1.0000 | 0.4528 | 0.4528 |
| Twitter | 0.9929 | 0.9911 | 0.9841 | 0.9306 | 0.9893 | 0.9206 | 0.9834 | 0.9381 | 0.9225 | 0.9970 | 0.1700 | 0.1695 | 0.9890 | 0.3300 | 0.3264 | 1.0000 | 0.5654 | 0.5654 |
| YouTube | 1.0000 | 0.9995 | 0.9995 | 0.9999 | 0.9987 | 0.9986 | 0.9993 | 0.9991 | 0.9984 | 0.9980 | 0.0700 | 0.0699 | 0.9820 | 0.7900 | 0.7758 | 1.0000 | 0.9572 | 0.9572 |
| Skitter | 0.9985 | 0.9997 | 0.9982 | 0.9857 | 0.9991 | 0.9848 | 0.9958 | 0.9970 | 0.9928 | 0.9990 | 0.1000 | 0.0999 | 0.9760 | 0.7000 | 0.6832 | – | – | – |

## G.2  Evaluation on Knapsack Constraint

In Table 4, we represent the performance of the pruning algorithms on Maximum Coverage under size constraint.

## G.3  Runtime

Table 5 reports the absolute inference runtimes for all evaluated algorithms on the Influence Maximization problem under the size constraint. All algorithms are evaluated using the same computational resources (identical numbers of threads and GPUs) to ensure a fair comparison. The only exception is SS, which requires additional threads on large graphs.

Table 4: Comparison of pruning algorithms for size-constraint experiments. Highlighted are the results ranked **first**, **second** , and **third**. *Values are reported from Nath & Kuhnle (2025).

| Dataset | QUICKPRUNE | | | TOP-K | | | LLM2PRUNE | | |
|---|---|---|---|---|---|---|---|---|---|
| | $P_r$ | $P_g$ | $C$ | $P_r$ | $P_g$ | $C$ | $P_r$ | $P_g$ | $C$ |
| **Maximum Coverage Weighted** | | | | | | | | | |
| Facebook | 0.9725 | 0.9871 | **0.9600** | 0.5505 | 0.9871 | **0.5434** | 0.9771 | 0.7501 | **0.7330** |
| DBLP | 0.8450 | 0.9997 | **0.8447** | 0.7150 | 0.9997 | **0.7148** | 1.0000 | 0.9984 | **0.9984** |
| Deezer | 0.8750 | 0.9983 | **0.8735** | 0.9150 | 0.9983 | **0.9134** | 1.0000 | 0.9907 | **0.9907** |
| Slashdot | 0.8950 | 0.9988 | **0.8939** | 0.5700 | 0.9988 | **0.5693** | 1.0000 | 0.9704 | **0.9704** |
| Wiki | 0.9400 | 0.9851 | **0.9260** | 0.7550 | 0.9851 | **0.7438** | 1.0000 | 0.8427 | **0.8427** |
| Twitter | 0.8700 | 0.9987 | **0.8689** | 0.5950 | 0.9987 | **0.5942** | 1.0000 | 0.9876 | **0.9876** |
| YouTube | 0.9150 | 0.9999 | **0.9149** | 0.5200 | 0.9999 | **0.5199** | 1.0000 | 0.9982 | **0.9982** |
| Skitter | 0.9050 | 0.9999 | **0.9049** | 0.8150 | 0.9999 | **0.8149** | 1.0000 | 0.9997 | **0.9997** |
| **Maximum Cut Weighted** | | | | | | | | | |
| Facebook | 0.9899 | 0.9666 | **0.9568** | 1.0000 | 0.9666 | **0.9666** | 1.0000 | 0.5002 | **0.5002** |
| DBLP | 0.9500 | 0.9999 | **0.9499** | 0.4300 | 0.9999 | **0.4300** | 0.9500 | 0.9984 | **0.9485** |
| Deezer | 0.9600 | 0.9985 | **0.9586** | 0.8000 | 0.9985 | **0.7988** | 0.9500 | 0.9907 | **0.9411** |
| Slashdot | 0.9700 | 0.9991 | **0.9691** | 0.6700 | 0.9991 | **0.6694** | 0.9500 | 0.9926 | **0.9430** |
| Wiki | 0.9600 | 0.9928 | **0.9531** | 0.5100 | 0.9928 | **0.5063** | 1.0000 | 0.9214 | **0.9214** |
| Twitter | 0.9500 | 0.9996 | **0.9496** | 0.2900 | 0.9996 | **0.2899** | 0.9500 | 0.9938 | **0.9441** |
| YouTube | 0.9700 | 0.9999 | **0.9699** | 0.7300 | 0.9999 | **0.7299** | 1.0000 | 0.9995 | **0.9995** |
| Skitter | 0.9700 | 0.9999 | **0.9699** | 1.0000 | 0.9999 | **0.9999** | 1.0000 | 0.9994 | **0.9994** |
| **Influence Maximization Weighted** | | | | | | | | | |
| Facebook | 0.9934 | 0.9148 | **0.9088** | 0.9934 | 0.9148 | **0.9088** | 1.0014 | 0.7501 | **0.7512** |
| DBLP | 1.0000 | 0.9958 | **0.9958** | 1.0000 | 0.9958 | **0.9958** | 0.9646 | 0.9937 | **0.9585** |
| Deezer | 1.0000 | 0.9820 | **0.9820** | 1.0000 | 0.9820 | **0.9820** | 0.9923 | 0.9813 | **0.9738** |
| Slashdot | 0.9132 | 0.9868 | **0.9011** | 1.0120 | 0.9868 | **0.9986** | 1.0067 | 0.9852 | **0.9918** |
| Wiki | 0.9660 | 0.8774 | **0.8476** | 1.0000 | 0.8774 | **0.8774** | 1.0098 | 0.6854 | **0.6922** |
| Twitter | 0.9121 | 0.9986 | **0.9108** | 0.9689 | 0.9986 | **0.9675** | 1.0005 | 0.9381 | **0.9385** |
| YouTube | 0.9171 | 0.9994 | **0.9165** | 0.9502 | 0.9994 | **0.9496** | 1.0400 | 0.9954 | **1.0353** |
| Skitter | 0.8291 | 0.9999 | **0.8290** | 0.9391 | 0.9999 | **0.9390** | 1.0345 | 0.9997 | **1.0342** |

LLM2PRUNE is 1-4 orders of magnitude faster than classical methods on larger graphs, completing inference in under 3 seconds even on Skitter.

Table 5: Absolute inference runtime (seconds) for Influence Maximization under size constraint. *SS uses additional threads on large graphs to speed up pruning; all other algorithms use the same number of threads and GPUs.

| Algorithm | Facebook | Wiki | Deezer | Slashdot | Twitter | DBLP | YouTube | Skitter |
|---|---|---|---|---|---|---|---|---|
| LLM2PRUNE | 0.194 | 0.203 | 0.235 | 0.257 | 0.314 | 0.458 | 1.118 | 2.420 |
| QUICKPRUNE | 1.860 | 41.960 | 1.100 | 226.000 | 3219.040 | 222.760 | 658.320 | 5109.000 |
| SS* | 24.960 | 86.424 | 11.371 | 102.259 | 1242.836 | 118.537 | 3652.793 | 10433.744 |
| GCOMB-P | 0.002 | 0.004 | 0.023 | 0.035 | 0.051 | 0.205 | 0.515 | 0.980 |
| LENSE | 31.453 | 36.476 | 42.164 | 44.193 | 81.610 | 49.286 | 238.906 | 1607.605 |

Table 6 breaks down the total inference time of LLM2PRUNE into feature extraction and GNN inference components.

Table 6: Runtime breakdown of LLM2PRUNE for Influence Maximization under size constraint.

| Dataset | Feature Extraction (s) | GNN Inference (s) | Total (s) |
|---------|----------------------|-------------------|-----------|
| Facebook | 0.006 | 0.188 | 0.194 |
| Wiki | 0.008 | 0.195 | 0.203 |
| Deezer | 0.058 | 0.177 | 0.235 |
| Slashdot | 0.063 | 0.193 | 0.257 |
| Twitter | 0.138 | 0.176 | 0.314 |
| DBLP | 0.284 | 0.175 | 0.458 |
| YouTube | 0.942 | 0.176 | 1.118 |
| Skitter | 2.195 | 0.225 | 2.420 |

### G.4  Computational Cost Breakdown

Table 7 provides a unified end-to-end computational cost breakdown for LLM2PRUNE, separating the one-time beam search cost from the per-dataset fine-tuning cost and the per-instance inference cost. The beam search requires approximately 24,000 LLM tokens per problem variant, reported in tokens since GPT-5-nano is closed-source. This one-time cost is amortized across all future instances of the same graph family. Per-dataset fine-tuning ranges from 1.98 to 36.35 GFLOPs depending on graph size, while per-instance inference costs only 2.61 to 558.09 MFLOPs, demonstrating that the amortized deployment cost of LLM2PRUNE is negligible relative to the one-time search cost.

Table 7: End-to-end computational cost breakdown for LLM2PRUNE. Beam search is a one-time cost per problem family, amortized across all instances. LLM cost is reported in tokens since GPT-5-nano is closed-source.

| Component | Scope | Cost |
|-----------|-------|------|
| *One-time beam search* | | |
| LLM ($\sim$65 calls per variant) | Per problem family | $\sim$24k tokens |
| GNN training (HK graph) | Per problem family | 108.00 GFLOPs |
| *Per-dataset fine-tuning (GFLOPs)* | | |
| Facebook | Per dataset | 1.98 |
| Wiki | Per dataset | 2.34 |
| DBLP | Per dataset | 11.34 |
| Slashdot | Per dataset | 14.44 |
| Twitter | Per dataset | 15.29 |
| Deezer | Per dataset | 15.11 |
| Skitter | Per dataset | 27.25 |
| YouTube | Per dataset | 36.35 |
| *Per-instance inference (MFLOPs)* | | |
| Facebook | Per instance | 2.61 |
| Wiki | Per instance | 3.15 |
| Deezer | Per instance | 17.69 |
| Slashdot | Per instance | 18.30 |
| Twitter | Per instance | 51.24 |
| DBLP | Per instance | 66.55 |
| YouTube | Per instance | 206.52 |
| Skitter | Per instance | 558.09 |

### G.5 Multi-budget Analysis.

In Table 8, we present the pruning approximation ratio ($P_r$) across a range of budgets under the size constraint following Ireland & Montana (2022). The results exhibit similar trends for the knapsack constraint.

Table 8: Performance of LLM2PRUNE across a range of budgets across three CO problems under size constraint.

| Maximum Coverage | | | | | | |
|---|---|---|---|---|---|---|
| **Dataset** | 1 | 10 | 25 | 50 | 75 | 100 |
| Facebook | 1.0000 | 1.0000 | 0.9982 | 0.9974 | 0.9959 | 0.9966 |
| Wiki | 1.0000 | 1.0000 | 1.0000 | 1.0000 | 0.9997 | 0.9988 |
| Deezer | 1.0000 | 1.0000 | 1.0000 | 0.9972 | 0.9962 | 0.9930 |
| Slashdot | 1.0000 | 1.0000 | 1.0000 | 0.9992 | 0.9963 | 0.9914 |
| Twitter | 1.0000 | 1.0000 | 1.0000 | 0.9993 | 0.9961 | 0.9834 |
| DBLP | 1.0000 | 1.0000 | 1.0000 | 1.0000 | 1.0000 | 1.0000 |
| YouTube | 1.0000 | 1.0000 | 1.0000 | 1.0000 | 1.0000 | 0.9993 |
| Skitter | 1.0000 | 1.0000 | 1.0000 | 1.0000 | 0.9974 | 0.9958 |
| **Maximum Cut** | | | | | | |
| **Dataset** | 1 | 10 | 25 | 50 | 75 | 100 |
| Facebook | 1.0000 | 1.0000 | 1.0000 | 1.0000 | 1.0004 | 1.0004 |
| Wiki | 1.0000 | 1.0000 | 1.0000 | 1.0000 | 1.0000 | 1.0000 |
| Deezer | 1.0000 | 1.0000 | 1.0000 | 1.0000 | 1.0000 | 0.9999 |
| Slashdot | 1.0000 | 1.0000 | 1.0000 | 1.0000 | 1.0000 | 1.0000 |
| Twitter | 1.0000 | 1.0000 | 1.0000 | 1.0000 | 1.0000 | 1.0000 |
| DBLP | 1.0000 | 1.0000 | 1.0000 | 0.9984 | 0.9979 | 0.9981 |
| YouTube | 1.0000 | 1.0000 | 1.0000 | 1.0000 | 1.0000 | 1.0000 |
| Skitter | 1.0000 | 1.0000 | 1.0000 | 1.0000 | 0.9994 | 0.9966 |
| **Influence Maximization** | | | | | | |
| **Dataset** | 1 | 10 | 25 | 50 | 75 | 100 |
| Facebook | 0.9802 | 0.9911 | 0.9905 | 0.9933 | 1.0035 | 1.0047 |
| Wiki | 0.9904 | 1.0181 | 1.0010 | 1.0057 | 1.0000 | 1.0014 |
| Deezer | 1.0015 | 1.0088 | 0.9738 | 0.9870 | 0.9784 | 0.9871 |
| Slashdot | 1.0225 | 1.0076 | 0.9967 | 1.0068 | 1.0079 | 1.0104 |
| Twitter | 0.9997 | 0.9937 | 0.9869 | 0.9819 | 0.9790 | 0.9695 |
| DBLP | 0.9490 | 0.9552 | 0.9653 | 0.9847 | 0.9903 | 0.9941 |
| YouTube | 0.9999 | 1.0036 | 0.9976 | 1.0082 | 1.0014 | 0.9995 |
| Skitter | 0.9998 | 0.9995 | 0.9992 | 0.9989 | 0.9991 | 1.0006 |

### G.6 Comparison of LLM2Prune with different LLMs

We evaluate the robustness of LLM2PRUNE with respect to the choice of the underlying LLM by comparing two representative LLMs: GPT-5-nano and LLaMA-3.3-70B-Instruct. Both models are used identically within the framework to propose features, generate executable code, and summarize feedback, while all downstream components, hyperparameters, and evaluation protocols are kept fixed.

Figure 6 reports the combined performance metric across all graph optimization tasks under size constraints. The results show that LLM2PRUNE achieves comparable performance with both LLMs, indicating that the framework is not sensitive to a specific model choice.

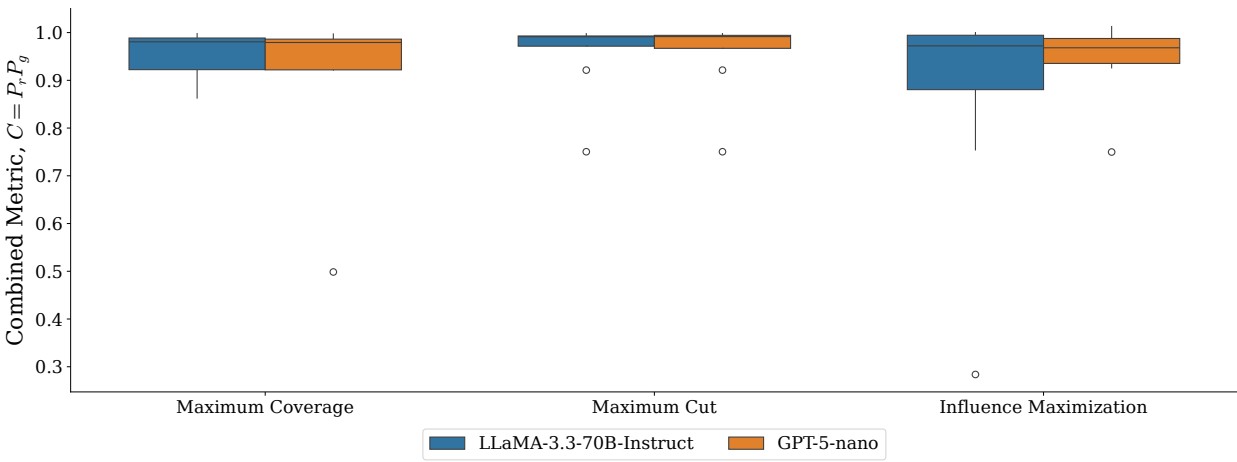

Figure 6: Comparison of LLM2PRUNE using different LLMs. The plot shows the combined performance metric across all graph optimization tasks under constraints. Both GPT-5-nano and LLaMA-3.3-70B-Instruct yield similar performance, demonstrating that LLM2PRUNE is robust to the choice of the underlying LLM.

