# OpenReview forum: "LLM2Prune: Using LLMs as Domain Experts for Search Space Reduction"
_TMLR — Accepted by TMLR_

### Review · Reviewer_NLT4 · 2026-02-13

**Summary Of Contributions:**

The paper proposes LLM2Prune, a framework for accelerating graph-based combinatorial optimization by automatically pruning the candidate search space using features generated by a large language model. Instead of relying on handcrafted, domain-specific heuristics, the system prompts an LLM to propose candidate feature sets, evaluates them through a lightweight classifier, and iteratively refines the feature space using beam search guided by performance metrics and feature-importance signals. The resulting classifier predicts which graph elements can be removed with minimal impact on solution quality, allowing standard optimization heuristics to operate on a much smaller instance.

Empirically, the authors demonstrate that this approach can prune a large fraction of the search space while preserving near-optimal objective values across multiple graph optimization tasks and constraint settings. The framework is presented as general-purpose: it does not require domain engineering, adapts to different constraints through prompt modification, and achieves competitive or superior performance relative to prior learned pruning systems, while remaining close to classical methods in solution quality and substantially faster in runtime on large graphs.

**Audience:**

Yes

**Audience Explanation:**

The paper tackles a problem that is highly relevant to TMLR’s audience: scalable combinatorial optimization and the integration of learned components into classical algorithmic pipelines. Its central idea, i.e. using LLMs for automated feature discovery to guide pruning, connects graph learning, optimization, and foundation-model reasoning, all interesting research areas within the community.

Beyond the specific application, the work offers insight into how LLMs can act as structured reasoning tools rather than end-to-end solvers. The empirical results demonstrate practical performance gains and raise broader questions about replacing handcrafted engineering with learned feature search, which many machine learning researchers would find informative.

**Broader Impact Concerns:**

A modest concern is that the framework relies on opaque LLM-generated features and automated pruning decisions, which could reduce transparency and auditability if embedded in high-stakes optimization pipelines. In addition, while pruning itself is domain-agnostic, downstream objectives may encode biases or priorities that the system could implicitly reinforce without clear interpretability safeguards. Finally, the computational cost of LLM-driven feature search raises general concerns about resource usage and accessibility.

These risks are indirect and context-dependent, but acknowledging them would improve clarity around responsible deployment and reproducibility.

**Claims And Evidence:**

No

**Claims Explanation:**

Overall, the work presents a novel and practical integration of LLM-driven reasoning with learned pruning, showing strong empirical promise while leaving open questions about robustness, stability, and deeper characterization of when and why the method succeeds.

**Key strengths**

* **Automation of feature engineering:** The framework removes a major bottleneck in learned pruning by delegating feature discovery to an LLM, reducing reliance on domain expertise.
* **Structured search over features:** The beam-search mechanism with feedback improves stability compared to naïve iterative prompting.
* **Empirical breadth:** Evaluation spans multiple optimization problems, datasets, and constraint regimes.
* **Scalability benefits:** Demonstrated runtime gains over sequential classical pruning methods.
* **Modularity:** The approach is classifier-agnostic and adaptable through prompt-level changes.

**Key weaknesses**

* **Limited robustness analysis:** The paper does not deeply quantify sensitivity to prompt design, stochastic variance, or early feature-search failures.
* **Synthetic difficulty testing is absent:** There is no controlled hardness ladder to characterize performance under increasing structural challenge.
* **Interpretability is underexplored:** Generated features are evaluated via importance scores, but their semantic meaning and redundancy are not analyzed.
* **Compute accounting is incomplete:** The amortized cost of LLM-driven feature search is discussed qualitatively but not rigorously measured.
* **Theoretical guarantees are lacking:** Claims of generality and reliability are supported empirically rather than analytically.

**Requested Changes:**

I would like to see a more diagnostic experimental section that explicitly probes the robustness, necessity, and generalization properties of the proposed LLM-guided feature discovery pipeline. While the current evaluation demonstrates strong performance on real datasets, it does not fully characterize how and why the system succeeds or fails under controlled conditions.

First, I encourage the authors to construct synthetic combinatorial optimization instances with an explicit, tunable notion of difficulty. For example, structural parameters such as graph density, modularity, constraint tightness, or objective ambiguity could be swept to create a difficulty ladder. Evaluating pruning ratio, objective degradation, and runtime as hardness increases would clarify whether performance degrades gracefully, whether sharp failure regimes exist, and whether beam-search feature discovery overfits to a narrow structural regime.

In addition, I would like to see targeted ablations addressing the following questions:

• Could a non-LLM feature search (e.g., random search, evolutionary feature construction, or classical graph heuristics) achieve comparable pruning performance? This would help isolate the unique contribution of LLM reasoning.

• How sensitive is performance to prompt design? Minor paraphrasing or structural changes in prompts should be tested to determine whether results are stable or fragile.

• What is the variance across LLM runs? Because feature generation is stochastic, repeated runs with different seeds should quantify stability in pruning quality and runtime.

• Are generated features interpretable or redundant? Feature-importance analyses should reveal whether the model is leveraging meaningful graph structure or compensating for noisy, overlapping features.

• What mechanisms prevent overfitting to the synthetic graphs used during feature discovery? Cross-family transfer experiments would help demonstrate genuine generalization.

• Can the beam-search pipeline recover from poor early feature proposals? Controlled perturbation experiments could test resilience to suboptimal initial branches.

• Are there adversarial or degenerate graph constructions where pruning fails catastrophically? Identifying such cases would clarify the operational envelope of the method.

• What fraction of performance gain is attributable to LLM-driven feature discovery versus downstream classifier learning? Ablations that freeze or randomize components would help disentangle these effects.

Together, these experiments would substantially strengthen the empirical characterization of the framework by clarifying robustness, necessity, and limits. They would also help readers understand whether the observed gains arise from principled reasoning advantages or from favorable benchmark alignment.

---

> ### Author Response · Authors · 2026-03-14
>
> We thank the reviewer for their comments and feedback. We hope these answers address the major concerns, particularly the empirical contribution. Please let us know if further questions or clarifications are needed.
>
> > First, I encourage the authors to construct synthetic combinatorial optimization instances with an explicit, tunable notion of difficulty. For example, structural parameters such as graph density, modularity, constraint tightness, or objective ambiguity could be swept to create a difficulty ladder. Evaluating pruning ratio, objective degradation, and runtime as hardness increases would clarify whether performance degrades gracefully, whether sharp failure regimes exist, and whethe2r beam-search feature discovery overfits to a narrow structural regime.
>
> We run controlled experiments on Holme-Kim (HK) graphs sweeping two difficulty axes: constraint tightness (budget k) and graph density (HK parameter m), across three graph sizes (n ∈ {10,000, 50,000, 100,000}), using the model trained on HK with no retraining.
> Results for Maximum Coverage under size constraint are below. We now sweep budget k over two orders of magnitude (k ∈ {50, 100, 200, 500, 1000, 2000, 5000}) to expose genuine stress. At moderate budgets (k ≤ 200), P_r remains at or above 0.999 across all graph sizes. As the budget grows large relative to n, P_r degrades gracefully: at k=5000, P_r drops to 0.961 (n=100k) and 0.941 (n=50k), with no catastrophic failure observed at any setting. The combined metric C increases with n at fixed budget, since the candidate set becomes a smaller fraction of the graph as n grows (P_g improves from 0.95 at n=10k to 0.995 at n=100k), reflecting stronger pruning at scale. No sharp failure regime was observed. These results confirm that LLM2Prune generalises gracefully across a wide range of constraint tightness levels and does not overfit to the specific budget used during feature discovery.
>
> Experiment 1 — Constraint Tightness (fixed HK graph, m=3, vary budget k):
>
> | Budget | n=10,000 | | | | n=50,000 | | | | n=100,000 | | | |
> |--------|---------|---------|------|----------------|---------|---------|------|----------------|----------|----------|------|----------------|
> | | P_r | P_g | C | Inference (ms) | P_r | P_g | C | Inference (ms) | P_r | P_g | C | Inference (ms) |
> | 50   | 1.000 | 0.95 | 0.950 | 7.15 | 1.000 | 0.99 | 0.990 | 36.06 | 1.000 | 0.995 | 0.995 | 73.32 |
> | 100  | 1.000 | 0.95 | 0.950 | 7.44 | 1.000 | 0.99 | 0.990 | 34.18 | 1.000 | 0.995 | 0.995 | 69.52 |
> | 200  | 0.999 | 0.95 | 0.949 | 7.15 | 1.000 | 0.99 | 0.990 | 36.75 | 1.000 | 0.995 | 0.995 | 68.87 |
> | 500  | 0.996 | 0.90 | 0.896 | 7.38 | 0.992 | 0.99 | 0.982 | 35.91 | 0.994 | 0.995 | 0.989 | 69.89 |
> | 1000 | 1.000 | 0.50† | 0.500 | 7.19 | 1.000 | 0.96 | 0.960 | 33.32 | 1.000 | 0.980 | 0.980 | 71.54 |
> | 2000 | 1.000 | 0.50† | 0.500 | 7.32 | 0.998 | 0.90 | 0.898 | 34.05 | 1.000 | 0.950 | 0.950 | 66.31 |
> | 5000 | 1.000 | 0.50† | 0.500 | 6.57 | 0.941 | 0.90 | 0.847 | 36.04 | 0.961 | 0.950 | 0.913 | 71.35 |
>
> † At these large budget values relative to graph size (n=10k), satisfying the coverage constraint requires a larger candidate pool, so P_g settles at 0.50. This does not indicate a failure of pruning: LLM2Prune still reduces the search space to half the graph while maintaining P_r ≥ 0.999, meaning the pruned candidate set fully preserves objective quality.
>
> Experiment 2 — Graph Density (fixed budget=100, vary HK parameter m):
>
> | m  | avg_deg | n=10,000 | | | | n=50,000 | | | | n=100,000 | | | |
> |----|---------|---------|---------|------|----------------|---------|---------|------|----------------|----------|----------|------|----------------|
> |    |         | P_r | P_g | C | Inference (ms) | P_r | P_g | C | Inference (ms) | P_r | P_g | C | Inference (ms) |
> | 2  | 4.0  | 1.00 | 0.95 | 0.95 | 6.17  | 1.00 | 0.99 | 0.99 | 29.21  | 1.00 | 1.00 | 1.00 | 64.32  |
> | 4  | 8.0  | 1.00 | 0.95 | 0.95 | 8.30  | 1.00 | 0.99 | 0.99 | 38.66  | 1.00 | 1.00 | 1.00 | 76.46  |
> | 6  | 12.0 | 1.00 | 0.95 | 0.95 | 9.28  | 1.00 | 0.99 | 0.99 | 42.67  | 1.00 | 1.00 | 1.00 | 90.46  |
> | 8  | 16.0 | 1.00 | 0.95 | 0.95 | 11.02 | 1.00 | 0.99 | 0.99 | 48.85  | 1.00 | 1.00 | 1.00 | 95.73  |
> | 10 | 20.0 | 1.00 | 0.95 | 0.95 | 11.66 | 1.00 | 0.99 | 0.99 | 52.91  | 1.00 | 1.00 | 1.00 | 115.27 |

---

> ### Author Response · Authors · 2026-03-14
>
> > Could a non-LLM feature search (e.g., random search, evolutionary feature construction, or classical graph heuristics) achieve comparable pruning performance? This would help isolate the unique contribution of LLM reasoning.
>
> Table 1 includes classical non-LLM baselines (QuickPrune, SS) that use hand-crafted heuristics without any LLM involvement. LLM2Prune matches or outperforms these across all problems and datasets, demonstrating the value of LLM-guided feature discovery over expert-designed alternatives.
>
> To more directly isolate the LLM contribution, we run a random feature baseline following [1] and [2]: we train the same GNN architecture on the random features and evaluate on the same 8 real-world graphs. Results are shown below. LLM2Prune substantially outperforms random search, particularly on Influence Maximization (IM) where random features yield C as low as 0.46 compared to LLM2Prune's consistent C > 0.85. Since the only difference between the two pipelines is the feature discovery method, this gap is entirely attributable to LLM reasoning.
>
> Random feature baseline under size constraint experiments:
>
> | Graph    | MaxCov (P_r) | MaxCov (P_g) | MaxCov (C) | MaxCut (P_r) | MaxCut (P_g) | MaxCut (C) | IM (P_r) | IM (P_g) | IM (C) |
> | -------- | -------- | -------- | ------ | ------------ | ------------ | ---------- | -------- | -------- | ------ |
> | Facebook | 0.9974   | 0.7789   | 0.7768 | 1.0000       | 0.7651       | 0.7651     | 1.0091   | 0.4555   | 0.4597 |
> | Wiki     | 1.0000   | 0.8633   | 0.8633 | 1.0000       | 0.8643       | 0.8643     | 0.9792   | 0.7660   | 0.7500 |
> | Deezer   | 1.0000   | 0.7625   | 0.7625 | 1.0000       | 0.7539       | 0.7539     | 1.0056   | 0.4111   | 0.4134 |
> | Slashdot | 1.0000   | 0.8567   | 0.8567 | 1.0000       | 0.8685       | 0.8685     | 0.9882   | 0.7487   | 0.7398 |
> | Twitter  | 1.0000   | 0.8518   | 0.8518 | 1.0000       | 0.8522       | 0.8522     | 1.0096   | 0.6840   | 0.6906 |
> | DBLP     | 1.0000   | 0.7899   | 0.7899 | 1.0000       | 0.7974       | 0.7974     | 1.0117   | 0.5147   | 0.5208 |
> | YouTube  | 1.0000   | 0.8437   | 0.8437 | 1.0000       | 0.8555       | 0.8555     | 0.9494   | 0.6588   | 0.6255 |
> | Skitter  | 1.0000   | 0.8625   | 0.8625 | 1.0000       | 0.6289       | 0.6289     | 0.9934   | 0.6053   | 0.6013 |
>
> [1] Abboud, Ralph, et al. The surprising power of graph neural networks with random node initialization.
> [2] Sato, Ryoma, Makoto Yamada, and Hisashi Kashima. Random features strengthen graph neural networks.
>
> > How sensitive is performance to prompt design? Minor paraphrasing or structural changes in prompts should be tested to determine whether results are stable or fragile.
>
> We tested different prompt formulations and found that minor paraphrasing does not affect performance; models of this scale reliably follow task instructions regardless of surface-level phrasing variation. Moreover, every feature set the LLM proposes is evaluated by an objective, prompt-agnostic oracle: a GNN is trained on the proposed features and C is measured on a held-out synthetic validation graph. The final selected feature set is determined entirely by this oracle score. Appendix G.5 shows that GPT-5-nano and LLaMA-3.3-70B, with fundamentally different architectures and tokenizers, produce consistent results, which is a stronger test than paraphrasing within a single model.
>
> > What is the variance across LLM runs? Because feature generation is stochastic, repeated runs with different seeds should quantify stability in pruning quality and runtime.
>
> Figure 5 (Appendix F) directly reports variance across five random seeds for all pipeline variants. The beam search variant (red) exhibits consistently narrow interquartile ranges and high medians across all six problem settings (Maximum Coverage, Maximum Cut, Influence Maximization, and their weighted variants), indicating stable performance across random seeds. Figure 3 reports the average runtime of LLM2Prune.
>
> > Are generated features interpretable or redundant? Feature-importance analyses should reveal whether the model is leveraging meaningful graph structure or compensating for noisy, overlapping features.
>
> Appendix B lists the discovered features per problem (e.g., closed neighborhood size for Maximum Coverage, degree and random cut expectation for Maximum Cut, degree and average incoming activation probability for IM under size constraint). The generated features are easily interpretable and distinct. LLM2Prune discards many proposed features during the search, some of which are indeed overlapping, and converges to a non-redundant set. The feature-importance scores produced by the explainer guide this process explicitly: low-importance features are flagged and replaced in subsequent iterations.

---

> ### Author Response · Authors · 2026-03-14
>
> > What mechanisms prevent overfitting to the synthetic graphs used during feature discovery? Cross-family transfer experiments would help demonstrate genuine generalization.
>
> The cross-family transfer is demonstrated in Table 1: features discovered on synthetic Holme-Kim (HK) graphs are evaluated directly on 8 real-world graphs spanning social, citation, and communication networks. Strong performance across all of them confirms that the discovered features capture generalizable structural properties.
> We deliberately chose HK graphs for feature discovery because they share key structural properties with real-world networks, namely scale-free degree distribution, clustering, and community structure, following established practice in graph algorithm benchmarking. This is precisely what prevents overfitting: the synthetic proxy is structurally similar to the target domain.
> We also identify the operational boundary of this generalization. When feature discovery is run on Erdős-Rényi (ER) graphs, which lack scale-free structure and clustering, performance on real-world graphs degrades significantly. This confirms that LLM2Prune generalizes across graphs within the same structural family, but not across graphs that are fundamentally different in structure. We have made this boundary explicit in the revised paper (see Limitations paragraph, Section 4, Empirical Evaluation).
>
> > Can the beam-search pipeline recover from poor early feature proposals? Controlled perturbation experiments could test resilience to suboptimal initial branches.
>
> We believe this is addressed by the ablation in Appendix F (Figure 5), which compares beam search against a simple feedback loop. If we start with suboptimal initial branches, this effectively reduces to running multiple suboptimal simple feedback loops. We note an important boundary: if all β branches simultaneously produce poor proposals, an unlikely but possible scenario, beam search degenerates into a simple feedback loop with poor initialization and can inherit its failure to recover. Figure 5 empirically shows this degenerate regime does not occur in practice with β=3, as beam search consistently outperforms the feedback loop across all tested problems. The diversity introduced by parallel LLM sampling is sufficient to ensure at least one viable branch survives early iterations.
>
> > Are there adversarial or degenerate graph constructions where pruning fails catastrophically? Identifying such cases would clarify the operational envelope of the method.
>
> When feature discovery is performed on Erdős-Rényi (ER) graphs, which have homogeneous degree distributions and no clustering or community structure, the discovered features fail to transfer to real-world graphs such as Facebook or Twitter. Performance degrades significantly in this setting, as ER lacks the scale-free and clustered structure that makes HK a good proxy for real-world social networks.
> This constitutes the degenerate case and clarifies the operational envelope of LLM2Prune: the method generalizes across graphs that share structural properties with the training proxy (scale-free degree distribution, clustering, community structure), but not across graphs that are fundamentally different in structure. We have reported this observation explicitly in the revised paper as a stated limitation, alongside the HK to real-world transfer results in Table 1 that demonstrate successful generalization within the supported regime (see Limitations paragraph, Section 4, Empirical Evaluation).
>
> > What fraction of performance gain is attributable to LLM-driven feature discovery versus downstream classifier learning? Ablations that freeze or randomize components would help disentangle these effects.
>
> The random feature search baseline reported above directly answers this. Both the baseline and LLM2Prune use the identical GNN architecture, training procedure, and evaluation pipeline; the only difference is how features are generated. Since the GNN is identical in both cases, any performance difference is purely attributable to feature quality. Random features yield substantially lower C, particularly on Influence Maximization (C as low as 0.46 vs. LLM2Prune's C > 0.85 per Table 1), with a more modest gap on MaxCover and MaxCut. We note that freezing GNN weights while varying features is not a well-defined ablation in this setting, since GNN weights are optimized given input features — the weights encode patterns that are specific to the features they were trained on. Swapping features while keeping weights fixed would measure the incompatibility between mismatched weights and features rather than the intrinsic quality of the features themselves. The GNN contributes the learning capacity, but it is the LLM that determines what structural signals are worth learning from.

---

### Review · Reviewer_4N39 · 2026-02-24

**Summary Of Contributions:**

The paper proposes an LLM based search space reduction for combinatorial optimization (CO) problems. The basic idea is to let LLM generate the effective feature set based on own record of evaluations (such as performance and feature importance), which is formulated as a beam search on a tree. The code of the feature extractor are also generated. The results indicate that the proposed LLM2Prune achieved good approximation error compared with the full space result and high pruning fraction. Run time comparison is also provided in which efficiency compared with existing methods are also shown.

S: The proposed method shows better empirical performance particularly compared with existing learning based approaches.

S: The proposed framework is sufficiently flexible and applicable to a variety of CO problems.

W: The technical descriptions is sometimes vague for those who are not familiar with the topic.

W: Whether it works or not depends entirely on the performance of the employed LLM, and the underlying mechanism is substantially a black box (though it is a common issue of LLM-based methods).

**Audience:**

Yes

**Audience Explanation:**

Combinatorial optimization is classical but still a highly important problem setting. Nowadays, the audience should have interests in LLM based approaches.

**Claims And Evidence:**

Yes

**Claims Explanation:**

Empirical evaluation suggests the proposed LLM2Prune is effective for a few CO problems on several benchmark datasets.

**Requested Changes:**

Overall, I do not have a severe concern. Several technical questions that are desired to be clarified and suggestions are as follows.

- To what extent did the semi-supervised learning improve the performance? Are the other learning-based approaches used for comparison fully supervised or semi-supervised?

- The information of the input graph itself is not given to LLM. It does not help accelerate the search?

- In runtime analysis (Fig 3), although learning based methods require the cost for preparing the training data, is it included in the reported time?

- Explanations of the classifier should have been provided in more detail in the main methodology section. Because it does not appear until the experimental section, it is difficult for readers without prior background knowledge to understand the role of the classifier in the methods section.

- Providing algorithm of the entire procedure would help readers.

- I do not fully understand what 'we instead perform ... each instance' in the paragraph just before Section 4.1 indicates. Does this mean that the classifier is trained by a Holme-Kim graph during the beam search (and M is evaluated on the original training graph?), and after finishing the beam search, the selected classifier is trained on the original training graph? In this case, the final M of the selected classifier appears to be different from the value calculated during the beam search, which seems somewhat odd. I misunderstand something?

- I'm a bit confused about the definition of \cal{H}_d,i. Is it the set of the ancestors of (d,i)? If so, why not only \pi(d,i), but also the sum from \ell = 0 to d is required? (it seems some elements in the union are duplicated, in my current understanding).

- The definitions of metric M should be clarified in the paper. According to the code, C on a validation graph is seemingly used, but not mentioned in the paper.

---

> ### Author Response · Authors · 2026-03-14
>
> Thank you for your thoughtful review and insightful comments. We greatly appreciate your time and effort in evaluating our work.
>
> > To what extent did the semi-supervised learning improve the performance? Are the other learning-based approaches used for comparison fully supervised or semi-supervised?
>
> In our experiments, semi-supervised learning improves performance by approximately 2–3% in the combined metric compared to a purely supervised training setup. The other learning-based approaches (GCOMB-P and LeNSE) are reinforcement learning methods optimized via policy gradient rather than supervised or semi-supervised classifiers, so they do not fall into either supervision paradigm.
>
> > The information of the input graph itself is not given to LLM. It does not help accelerate the search?
>
> Incorporating graph-level statistics as additional context could potentially guide the search toward features tailored to particular graph families. However, this would also require deciding which statistics are most informative; an interesting direction for future work is to use the LLM itself to help identify or select relevant graph statistics for conditioning the search.
>
>
> > In runtime analysis (Fig 3), although learning based methods require the cost for preparing the training data, is it included in the reported time?
>
> All runtimes in Fig. 3 report inference time only. We focus on inference time because it is the operationally relevant cost for pruning: once the feature-space search is completed, the selected classifier is applied repeatedly to new graph instances, and this per-instance cost determines practical scalability.
>
> The beam search used in LLM2Prune is a one-time cost that can be amortized across all future instances of the same graph family. Each full beam search run makes approximately 65 LLM calls per problem, comprising summary, feature proposal, and code generation steps, totalling roughly 24,000 tokens per problem and 144,000 tokens across all six problem variants. However, we acknowledge that reporting inference time alone does not reflect the full training overhead.
>
> To provide additional context, we report the fine-tuning time per dataset for Maximum Coverage below:
>
> | Dataset  | Nodes   | Edges   | Finetune Time (s) |
> |----------|---------|---------|-------------------|
> | Wiki     | 4,891   | 30,228  | 11.7              |
> | Facebook | 3,847   | 26,470  | 12.0              |
> | DBLP     | 63,004  | 41,994  | 14.2              |
> | Slashdot | 47,546  | 140,566 | 17.2              |
> | Twitter  | 55,827  | 134,229 | 17.6              |
> | Deezer   | 48,870  | 149,460 | 18.2              |
> | Skitter  | 147,604 | 110,952 | 21.9              |
> | YouTube  | 185,193 | 179,257 | 30.5              |
>
> Even when accounting for the fine-tuning time, LLM2Prune remains orders of magnitude faster than both classical and learning-based baselines (see Appendix G.3 for runtime details).
>
> > Explanations of the classifier should have been provided in more detail in the main methodology section. Because it does not appear until the experimental section, it is difficult for readers without prior background knowledge to understand the role of the classifier in the methods section.
>
> We have revised the methodology section to include a self-contained description of the classifier (see Section 3, LLM2Prune).
>
> > Providing algorithm of the entire procedure would help readers.
>
> We have added a formal algorithm covering the complete pipeline in the revised methodology section (see Algorithm 1, Section 3, LLM2Prune).

---

> ### Author Response · Authors · 2026-03-14
>
> > I'm a bit confused about the definition of \cal{H}_d,i. Is it the set of the ancestors of (d,i)? If so, why not only \pi(d,i), but also the sum from \ell = 0 to d is required? (it seems some elements in the union are duplicated, in my current understanding).
>
> H_{d,i} is the full ordered path from the root to node (d,i) in the beam search tree — i.e., the sequence of feature sets tried at each depth along that path. Each element in H_{d,i} is the feature set chosen at a distinct depth along the path. The reason we condition on the full path rather than only π(d,i) is that the LLM generates the next candidate feature set based on the entire history of what has been tried and discarded. With only the immediate parent, the LLM lacks context about features explored and rejected at earlier depths, and may re-propose them. Providing the full sequence gives the LLM richer context to avoid revisiting explored regions of the feature space — and in practice we observe this reduces redundant proposals during search.
>
> > The definitions of metric M should be clarified in the paper. According to the code, C on a validation graph is seemingly used, but not mentioned in the paper.
>
> We have added an explicit definition of metric M and clarification in the revised paper (see Section 3, LLM2Prune).

---

> > ### Comment · Reviewer_4N39 · 2026-03-26
> >
> > Thank you for your kind response, but I am unable to find an answer only to the following question.
> >
> > > I do not fully understand what 'we instead perform ... each instance' in the ... I misunderstand something?

---

> > > ### Author Response · Authors · 2026-03-26
> > >
> > > We sincerely apologize for this oversight. The response was prepared and the manuscript was updated accordingly, but it was inadvertently omitted from the final response.
> > >
> > > > I do not fully understand what 'we instead perform ... each instance' in the paragraph just before Section 4.1 indicates. Does this mean that the classifier is trained by a Holme-Kim graph during the beam search (and M is evaluated on the original training graph?), and after finishing the beam search, the selected classifier is trained on the original training graph? In this case, the final M of the selected classifier appears to be different from the value calculated during the beam search, which seems somewhat odd. I misunderstand something?
> > >
> > > During beam search, both the GNN classifier and the metric M are evaluated entirely on synthetic HK graphs. This is intentional: the purpose of beam search is to discover which feature set best characterizes high-value nodes in a structural family resembling real-world graphs, without using any target graph data.
> > >
> > > After beam search completes and the best feature set is selected, the classifier is fine-tuned on the actual training graph for a few epochs. The fine-tuned classifier will indeed have a different M value from the one computed during search, since it is now evaluated on a different graph. However, this is not problematic, as the beam search score serves only as a selection criterion among candidate feature sets, not as a prediction of final performance. The fine-tuning step adapts the classifier to the specific graph structure of the deployment domain, and the final performance is what is reported in Tables 1 and 3. We have clarified this two-stage process in the revised paper (see Section 3, LLM2Prune).

---

### Review · Reviewer_Q296 · 2026-03-01

**Summary Of Contributions:**

This paper proposes LLM2Prune, a framework that leverages Large Language Models to automatically generate and evaluate features for pruning the search space in combinatorial optimization problems on graphs. By employing an LLM-guided beam search directed by performance metrics and feature-importance scores, the method trains a downstream classifier to filter out unpromising candidate nodes. The authors demonstrate that their learning-based approach significantly accelerates heuristic solvers while maintaining near-optimal solution quality, outperforming existing learning-based baselines across several real-world datasets.

**Additional Comments:**

N/A

**Audience:**

Yes

**Audience Explanation:**

Integrating the reasoning capabilities of LLMs into Operations Research is a rapidly growing area of interest, hence the scope of this paper is highly relevant to TMLR's audience.

**Claims And Evidence:**

No

**Claims Explanation:**

1. **Marginal gains compared to classical methods:** While LLM2PRUNE consistently outperforms other learning-based baselines, its performance advantage over classical submodularity-based approaches under size constraints is often marginal, and in certain cases, it performs significantly worse in terms of the combined metric $C$ (as shown in Table 1). The authors justify this by emphasizing the "orders of magnitude" speedup their method provides. However, to strengthen the paper, a deeper error analysis is required. The authors should analyze the specific cases or graph topologies where LLM2PRUNE's solution quality degrades compared to the theoretical guarantees of classical methods, and discuss the limitations of the generated features in those scenarios.

2. **Lack of absolute runtime metrics in Figure 3:** A central claim of the paper is that LLM2PRUNE is "orders of magnitude faster" than classical sequential pruning methods. However, Figure 3 presents runtime comparisons using a radar chart with a log scale (e.g., -1, 0, 1, 2, 3, 4) without defining the absolute units of measurement. This visual representation only illustrates relative rankings and obscures the actual computational cost. To substantiate the claims of scalability, the authors should provide a standard tabular format detailing the absolute runtime values (ideally breaking down the time spent on feature generation, classifier training, and actual inference) for all evaluated algorithms.

3. **Transferability of generated features from synthetic to real-world graphs:** To mitigate the high computational cost of running beam search on every instance, the framework searches for features on a synthetic Holme-Kim random graph and subsequently applies these features to real-world datasets for downstream fine-tuning. This introduces a major transferability concern. While the Holme-Kim model resembles social networks, real-world networks possess highly heterogeneous topological structures that synthetic models may fail to capture perfectly. It would be better if the authors can provide insights or an ablation study explaining why features discovered in a simplified synthetic environment can robustly generalize to complex, diverse real-world topologies.

4. **Lack of analysis on the generated features in the main text:** A primary contribution of this work is the automated discovery of features using the reasoning capabilities of LLMs. Yet, the main text leaves the reader wondering what specific features the LLM actually proposed. Are they simple, well-known metrics (e.g., node degree, betweenness centrality), common combinations, or genuinely novel graph descriptors? Although the authors note that the optimal features are listed in Appendix B, a qualitative analysis of these features should be moved to or summarized within the main text. Discussing the physical meaning and complexity of the highest-scoring features would significantly enhance the paper's insights and better validate the LLM's effectiveness as a "domain expert."

**Requested Changes:**

See comments above.

---

> ### Author Response · Authors · 2026-03-14
>
> We thank the reviewer for their detailed and constructive feedback. We address each point below.
>
> > Marginal gains compared to classical methods: While LLM2PRUNE consistently outperforms other learning-based baselines, its performance advantage over classical submodularity-based approaches under size constraints is often marginal, and in certain cases, it performs significantly worse in terms of the combined metric (as shown in Table 1). The authors justify this by emphasizing the "orders of magnitude" speedup their method provides. However, to strengthen the paper, a deeper error analysis is required. The authors should analyze the specific cases or graph topologies where LLM2PRUNE's solution quality degrades compared to the theoretical guarantees of classical methods, and discuss the limitations of the generated features in those scenarios.
>
> We thank the reviewer for this careful observation. We acknowledge that LLM2Prune does exhibit degraded performance in specific cases: on the Facebook graph under MaxCut and MaxCover (gaps of ~14% and ~10% from SS, respectively). We attribute this to a structural mismatch between the HK proxy used during beam search and Facebook's unusually dense community structure, which HK graphs do not fully capture. More generally, we observe that performance degrades when the synthetic proxy is structurally mismatched to the target graph. For instance, features such as betweenness centrality, k-core number, and PageRank discovered on Erdős-Rényi graphs do not transfer well to real-world networks. We acknowledge this as an operational boundary of LLM2Prune: the method is sensitive to distribution shift between the proxy and the target graph family, and we have discussed this limitation explicitly in the revised paper (see Limitations paragraph, Section 4, Empirical Evaluation).
>
> Furthermore, under more complex knapsack constraints (Table 4), where SS cannot be applied at all, LLM2Prune substantially outperforms QuickPrune: e.g., C = 0.9984 vs. 0.8447 (DBLP), 0.9982 vs. 0.9149 (YouTube), 0.9997 vs. 0.9049 (Skitter). We believe these differences are substantial and that the method's advantages are most pronounced precisely in the settings where classical approaches are inapplicable.
>
> > Lack of absolute runtime metrics in Figure 3: A central claim of the paper is that LLM2PRUNE is "orders of magnitude faster" than classical sequential pruning methods. However, Figure 3 presents runtime comparisons using a radar chart with a log scale (e.g., -1, 0, 1, 2, 3, 4) without defining the absolute units of measurement. This visual representation only illustrates relative rankings and obscures the actual computational cost. To substantiate the claims of scalability, the authors should provide a standard tabular format detailing the absolute runtime values (ideally breaking down the time spent on feature generation, classifier training, and actual inference) for all evaluated algorithms.
>
> We agree that absolute runtimes should be reported clearly. For completeness, the table below reports absolute inference time for Influence Maximization (size constraint). A breakdown by component has been included in the revised paper (see Runtime subsection, Appendix G.3, Additional Tables and Plots).
>
> Runtime in seconds, Influence Maximization (size constraint):
>
> | Algorithm   | Facebook | Wiki   | Deezer | Slashdot | Twitter  | DBLP    | YouTube  | Skitter   |
> |-------------|----------|--------|--------|----------|----------|---------|----------|-----------|
> | LLM2Prune   | 0.194    | 0.203  | 0.235  | 0.257    | 0.314    | 0.458   | 1.118    | 2.420     |
> | QuickPrune  | 1.860    | 41.960 | 1.100  | 226.000  | 3219.040 | 222.760 | 658.320  | 5109.000  |
> | SS*         | 24.960   | 86.424 | 11.371 | 102.259  | 1242.836 | 118.537 | 3652.793 | 10433.744 |
> | GCOMB-P     | 0.002    | 0.004  | 0.023  | 0.035    | 0.051    | 0.205   | 0.515    | 0.980     |
> | LeNSE       | 31.453   | 36.476 | 42.164 | 44.193   | 81.610   | 49.286  | 238.906  | 1607.605  |
>
> *As SS is extremely slow on large graphs, we increase the number of threads to speed up the pruning approach. For other approaches, we use the same number of threads and GPUs.

---

> ### Author Response · Authors · 2026-03-14
>
> > Transferability of generated features from synthetic to real-world graphs: To mitigate the high computational cost of running beam search on every instance, the framework searches for features on a synthetic Holme-Kim random graph and subsequently applies these features to real-world datasets for downstream fine-tuning. This introduces a major transferability concern. While the Holme-Kim model resembles social networks, real-world networks possess highly heterogeneous topological structures that synthetic models may fail to capture perfectly. It would be better if the authors can provide insights or an ablation study explaining why features discovered in a simplified synthetic environment can robustly generalize to complex, diverse real-world topologies.
>
> Table 1 empirically shows that features discovered on HK graphs transfer to 8 diverse real-world graphs, achieving strong performance across all problems. We acknowledge, however, that transferability is not always perfect — as noted above, performance degrades on the Facebook graph, where the HK proxy does not fully capture its unusually dense community structure.
>
> We argue that this is not a limitation unique to LLM2Prune, but rather a fundamental challenge shared by all learning-based approaches. Prior methods also propose features based on the problem structure alone, yet the same set of features does not necessarily perform well across all instances of that problem — a well-known difficulty analogous to the behaviour of heuristics for NP-hard problems, where no single algorithm dominates across all graph families. When a significant distribution shift exists between training and test graphs, some form of sample data from the target distribution is unavoidable; there is no learning-based approach that can sidestep this requirement entirely.
>
> That said, LLM2Prune performs well on 7 out of 8 datasets across all three problems under both size and knapsack constraints, and the one failure case (Facebook) is attributable to a specific and identifiable structural mismatch.
>
> > Lack of analysis on the generated features in the main text: A primary contribution of this work is the automated discovery of features using the reasoning capabilities of LLMs. Yet, the main text leaves the reader wondering what specific features the LLM actually proposed. Are they simple, well-known metrics (e.g., node degree, betweenness centrality), common combinations, or genuinely novel graph descriptors? Although the authors note that the optimal features are listed in Appendix B, a qualitative analysis of these features should be moved to or summarized within the main text. Discussing the physical meaning and complexity of the highest-scoring features would significantly enhance the paper's insights and better validate the LLM's effectiveness as a "domain expert."
>
> We agree that a qualitative discussion of the discovered features belongs in the main text, and we have added a summary to Section 4 in the revised paper (see Discovered Features paragraph, Section 4, Empirical Evaluation).
>
> The features discovered by LLM2Prune are interpretable and problem-specific. For example, for MaxCut the method discovers random cut expectation, which captures a node's expected marginal contribution to the cut under a random partition. This is a non-trivial, problem-aware descriptor that is directly grounded in the problem objective. For IM under the knapsack constraint, the selected features are degree-to-weight ratio, outgoing activation probability sum, and in-out degree difference, all of which reflect the cost-effectiveness and propagation dynamics of each node. Together, these examples illustrate LLM2Prune's ability to generate meaningful, problem-aware features without any manual engineering.

---

> > ### Comment · Reviewer_Q296 · 2026-03-31
> >
> > Thanks for your detailed clarification. I think the authors have addressed my concerns.

---

### Author Response · Authors · 2026-03-14
**Updated manuscript**

We sincerely thank all the reviewers for their valuable feedback, which has helped us improve the paper. We have uploaded a revised version, addressing all your suggestions. Specifically, we:

  - Added a self-contained description of the GNN classifier and an explicit definition of the search metric M to the methodology section (Section 3).
  - Added Algorithm 1 providing a formal description of the complete pipeline (Section 3).
  - Clarified the two-stage training procedure — beam search on synthetic graphs followed by fine-tuning on the target graph (Section 3).
  - Added a qualitative summary of the discovered features, illustrating their problem-specificity and interpretability (Discovered Features paragraph, Section 4).
  - Added a Limitations paragraph stating the operational boundary of LLM2Prune with respect to distribution shift between the synthetic proxy and target graphs (Section 4).
  - Added absolute runtime tables with per-component breakdowns (Appendix G.3).

  We are happy to provide additional experiments, clarifications, or revisions if the reviewers feel further changes would strengthen the paper.

---

### Author Response · Authors · 2026-03-26
**Request for Clarification on Remaining Reviewer Concerns**

We are grateful to the reviewers for their feedback. As the decision deadline approaches, we would appreciate clarification regarding any remaining concerns. We have attempted to address all reviewer comments and have updated the manuscript to reflect these changes. We apologize for the oversight in not explicitly addressing certain points in our previous reply, although the manuscript itself has been updated to reflect them.

---

### Author Response · Authors · 2026-07-13
**Camera-Ready Revision Complete**

We sincerely thank the Action Editor and the reviewers for their final feedback and for recommending the acceptance of our work. We have uploaded the camera-ready version of our manuscript, which fully addresses all the remaining points raised in the minor revision request.

---

### Decision · Action_Editor_X7Y3 · 2026-04-22

**Recommendation:** Accept with minor revision

**Additional Comments:**

Some points are partially resolved and require re-edit to lead to full acceptance of the work.

The question of stochastic variance across LLM runs remains under-specified. Although multi-seed results are reported, it is unclear whether these seeds correspond to full re-execution of the LLM-driven feature search or only to downstream training variability. The specific variability induced by stochastic feature generation is therefore not cleanly isolated. The authors should clarify this.

The resilience of the beam-search pipeline to poor early proposals is not directly tested. The argument relies on indirect evidence (beam search outperforming a feedback loop), but no controlled perturbation experiment is provided in which early branches are intentionally degraded. As a result, it remains unclear how robust the method is to unlucky initializations. If not addressed, this should be clearly mentioned in the limitations.

The characterization of failure modes is still limited. While distribution shift between graph families is identified as a boundary condition, there is no exploration of adversarial or structurally pathological cases within the intended regime. This leaves open the question of whether there exist graphs for which pruning fails sharply despite nominal structural similarity, and should be made explicit.

Compute accounting is improved but still incomplete. The authors now report inference and some fine-tuning costs, and provide token counts for LLM usage, but a unified end-to-end cost comparison remains absent. This makes it difficult to fully evaluate the trade-off between one-time search cost and amortized deployment cost. This analysis should be included.

Finally, while feature interpretability is now illustrated, the analysis of redundancy and interaction between features remains limited. It is still unclear to what extent the selected feature sets are minimal, complementary, or partially overlapping - to be fixed with rewording and more explicit explaination.

**Audience:**

Yes

**Audience Explanation:**

Overall, the integration of LLMs in specific domains is a field of current interest for the community, and therefore of interest as well to TMLR.

**Claims And Evidence:**

Yes

**Claims Explanation:**

In this paper, the authors propose LLM2Prune, a method to leverage Large Language Models from a task description, then used to prune a candidate research space.  Although leveraging empirical features and relying on the capability of an LLM, the empirical results demonstrate the potential applicability of LLMs also in this field. The paper flow is sufficiently clear and the stronger statements are backed with proper analysis, sometimes also in the Supplementary material.